# Simple is Better than Complex: A Representation-centric Perspective for Prompting-based Vision–Language Fusion

## Abstract

Interactive prompting is an appealing approach to vision–language fusion using frozen unimodal transformers, yet recent progress often relies on increasingly complex prompting architectures. A natural question arises: instead of refining prompt designs, can fusion be improved more effectively by directly adapting internal representations within attention layers? Our analysis, from a representation-centric perspective, suggests that within each frozen attention layer, prompt tokens have limited direct control over the value representations of original modality tokens and their query–key interactions, motivating a lightweight alternative that targets these internal attention representations rather than increasing prompting complexity. Specifically, we use this analysis to guide where lightweight adaptation is applied: we investigate the cross-attention mechanism and propose combining value-only low-rank adaptation with a key–query replacement strategy, yielding a simple and parameter-efficient fusion design. Across common multimodal fusion benchmarks, the proposed method consistently outperforms prior prompting-based fusion baselines while requiring fewer trainable parameters. These results, along with further ablations, support representation-centric adaptation as an effective principle for prompting-based vision–language fusion in the frozen-encoder setting.

## 1 Introduction

Recent advances in large-scale pre-trained language models (Devlin et al., 2019; Radford et al., 2018) and vision models (Dosovitskiy, 2020; Liu et al., 2021) have stimulated growing interest in combining such unimodal architectures for vision–language tasks (Radford et al., 2021; Zhou et al., 2022b; Yang et al., 2022; khattak et al., 2023; Kiela et al., 2019; Yang et al., 2024). Early multimodal approaches typically trained the large unimodal encoders jointly with fusion modules (Hori et al., 2017; Kiela et al., 2019; Han et al., 2022). However, as these pre-trained architectures continue to grow in size, full fine-tuning becomes increasingly expensive, especially in multimodal settings where labeled data are often limited.

The demand for *sample efficiency* has therefore motivated a parallel line of work on parameter-efficient fine-tuning (PEFT; Lester et al., 2021; Li & Liang, 2021), including multimodal prompt tuning (Nagrani et al., 2021). These methods freeze the large unimodal backbones and instead tune a small set of prompt tokens. Starting from attention bottlenecks for multimodal fusion (Nagrani et al., 2021), which appended shared prompt tokens to unimodal token sequences to capture cross-modal interactions, subsequent work has sought to improve fusion by employing increasingly expressive prompting architectures. For example, Li et al. (2023) divided prompt tokens into different roles and projected part of them across modalities via a mapping function, while Jiang et al. (2024) further introduced a Mixture-of-Experts (MoE) architecture to generate longer and more adaptive prompts. While effective, these designs also introduce additional complexity in both architecture and optimization, raising a natural question: can multimodal fusion be improved more directly by adapting representations, rather than by continually refining prompting architectures?

This question is also consistent with a broader trend in multimodal learning. Recent work therein has increasingly explored direct tuning or alignment of internal multimodal representations (Liu et al., 2025b; Yoon et al., 2025), and contrastive-learning-based methods likewise highlight the importance of representa-

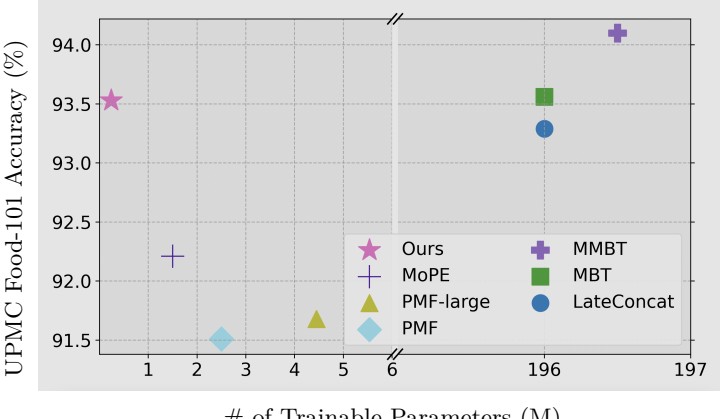

Figure 1: **Comparison of prompt-tuning and fine-tuning methods** on UPMC Food-101 in terms of classification accuracy and trainable parameter count (in millions). Among the prompt-tuning methods, our method achieves the best accuracy, and it approaches the performance of the strong full fine-tuning baselines with far fewer trainable parameters.

tion quality and cross-modal alignment (Radford et al., 2021; khattak et al., 2023; Chowdhury et al., 2023). Recent evidence further suggests that prompt tuning studies have paid limited attention to the evolution of internal attention representations during adaptation (Li et al., 2026).

In particular, we study this question by revisiting prior analyses of prompt tuning in language models (He et al., 2022; Chen et al., 2022b) and examining their implications in multimodal prompt tuning. Our analysis suggests that prior prompt-centric fusion designs mainly improve cross-modal fusion by strengthening prompt tokens. However, within a frozen attention layer, these tokens do not directly adapt the value projection applied to original modality tokens, nor do they directly change the pairwise query–key interactions among original modality tokens, thus limiting their ability to steer the resulting multimodal representations (c.f. Section 5.3 for empirical evidence). This leads us to a representation-centric alternative that strengthens the internal modality token representation in attention while retaining high parameter efficiency. As previewed in Figure 1, the resulting method achieves stronger performance than prior prompting-based approaches and remains competitive with heavier full fine-tuning baselines. In summary, we highlight our contributions as:

- We revisit prompting-based vision–language fusion from a representation-centric perspective and analyze the limitations of improving fusion solely through increasingly complex prompt designs in the frozen-encoder setting.

- We propose a parameter-efficient fusion method that combines targeted representation adaptation via value-only LoRA adaptation with a lightweight key–query replacement strategy.

- We conduct extensive experiments on multiple benchmark datasets and show that our method consistently outperforms prior prompting-based baselines while using fewer trainable parameters.

## 2   Related Works

In this section, we briefly introduce previous studies on multimodal fusion and then delve into prompting-based methods, especially those concerning vision–language fusion.

**Multimodal fusion.**   Multimodal fusion methods aim to simultaneously integrate and process signals from multiple modalities. Representative applications include: vision and language reasoning (Agrawal et al., 2015), sentiment analysis (Zadeh et al., 2016; Gandhi et al., 2022), medical analysis (Baños et al., 2015; Wang et al., 2021), detection and segmentation (Song et al., 2015; 2019), etc.

Very roughly, fusion can be categorized into early fusion, intermediate fusion, and late fusion, depending on where fusion is conducted (Atrey et al., 2010). Among these, late fusion remains widely used because of its interpretability and relative simplicity (Snoek et al., 2005; Zhang et al., 2023), and various improvements have been made. Some attempts mathematically robustify late fusion through rank minimization (Ye et al., 2012) and majority votes (Morvant et al., 2014). Beyond such late-fusion formulations, many recent methods design more expressive fusion modules, including hierarchical fusion architectures (Zhou et al., 2023) and memory-augmented recurrent fusion mechanisms (Islam et al., 2023). Moreover, several studies emphasize that multimodal fusion benefits from exploiting selected informative features or modalities, rather than treating all modalities equally for every sample (Hori et al., 2017; Han et al., 2020; 2022; Xue & Marculescu, 2023; Ni et al., 2023; Liu et al., 2023). In this paper, we focus on vision–language fusion with pre-trained unimodal transformers via interactive prompting, where *intermediate fusion* is heavily studied; we detail more in the next part.

**Prompting-based fine-tuning.** With the recent success of large-scale pre-trained unimodal transformers (Devlin et al., 2019; Dosovitskiy, 2020; Liu et al., 2021; Touvron et al., 2023), many works study how to adapt such architectures to downstream tasks through prompting. For **unimodal tasks**, early attempts leverage prompt tuning to efficiently adapt pre-trained language transformers for unimodal natural language processing (NLP; Lester et al. 2021; Li & Liang 2021). Prompt learning has also been widely studied on contrastively pre-trained vision–language models such as CLIP (Radford et al., 2021), including largely unimodal downstream settings such as image classification (Zhou et al., 2022b; khattak et al., 2023). The primary focus of this work is on integrating unimodally pre-trained transformers for **multimodal tasks**.

Some existing methods instead rely on multimodally pre-trained models or on fusion-based vision–language pre-training. For example, CoCoOp (Zhou et al., 2022a) learns instance-conditional text prompts for CLIP, while MaPLe (khattak et al., 2023) extends prompt learning to both the vision and language branches of CLIP through cross-branch prompt coupling. SDPT (Zhou et al., 2024) targets fusion-based vision–language pre-trained models and introduces shared prototype prompts with fixed inverse projections for different modality branches. These methods are related in spirit, but they are built on different pre-training and architectural settings of this work, and are therefore not the most direct comparison.

Back to the fusion of unimodally pre-trained transformers, some works utilize an extra encoder to map vision outputs to the input space of the text transformer for concatenation (Kiela et al., 2019; Tsimpoukelli et al., 2021; Mañas et al., 2022). There are also parameter-efficient transfer learning methods built on adapters rather than prompt tokens (Sung et al., 2022; Chowdhury et al., 2023; Seputis et al., 2024; Lu et al., 2023), which insert lightweight trainable modules into the backbone while keeping most backbone weights frozen. These methods are also related, but they differ from prompting-based interaction between otherwise frozen unimodal transformers.

Our work is in line with Nagrani et al. (2021); Liang et al. (2022); Li et al. (2023); Jiang et al. (2024). A dual-transformer framework is considered therein, where each transformer only processes a single modality and remains frozen; a sequence of additional prompt tokens is learned for information exchange between the two modalities.

## 3 Preliminaries

We introduce and formulate a few preliminaries in this section. We start with transformers in multimodal tasks, followed by the formulation of interactive prompting, and lastly low-rank adaption to be used in our approach.

### 3.1 Multimodal transformers

We formulate the forward pass of multimodal transformers for two modalities, i.e., text and image in our context (we defer full formulation to Appendix C.1). We denote the input sequence $\mathbf{z}$ as

$$\mathbf{z} = [z_{\mathrm{cls}}, \mathbf{E}(x_1), \mathbf{E}(x_2), \ldots, \mathbf{E}(x_N)], \tag{1}$$

where $\mathbf{E}(\cdot)$ is a mapping function to map the input token $x_i$ to $\mathbb{R}^d$, $z_{\text{cls}}$ is a special classification token prepended to the sequence. The input $\mathbf{z}$ are then passed to $L$ transformer layers, each consisting of a multi-head self-attention (MHA) module and a multilayer perceptron (MLP) block. Reloading the input sequence defined in Equation (1) as $\mathbf{z}^0$, we can repeatedly compute the output of the $(l+1)$-th transformer layer $\mathbf{z}^{l+1} = \mathbf{Transformer}(\mathbf{z}^l)$ as

$$
\begin{aligned}
\mathbf{y}^{l+1} &= \text{LN}\left(\text{MHA}^{l+1}(\mathbf{z}^l)\mathbf{W}_O^{l+1}\right) + \mathbf{z}^l, \\
\mathbf{z}^{l+1} &= \text{MLP}^{l+1}(\mathbf{y}^{l+1}) + \mathbf{y}^{l+1},
\end{aligned}
\tag{2}
$$

where LN denotes the layer normalization; (for simplicity) we illustrate the mechanism of $\text{MHA}^{l+1}(\mathbf{z})$ in a single-head form,

$$
\text{Attn}\left(\mathbf{z}\mathbf{W}_Q^{l+1}, \mathbf{z}\mathbf{W}_K^{l+1}, \mathbf{z}\mathbf{W}_V^{l+1}\right),
$$

where $\text{Attn}(\mathbf{Q}, \mathbf{K}, \mathbf{V}) = \text{softmax}\left(\mathbf{Q}\mathbf{K}^\top/\sqrt{d}\right)\mathbf{V}$ with sets of trainable parameters $\mathbf{W}_{\{Q,K,V,O\}} \in \mathbb{R}^{d \times d}$. The above procedure is performed in both modalities, and the final prediction, in a certain manner, combines the outputs of the two classification tokens from the two modalities.

### 3.2 Prompting-based attention bottlenecks

Above multimodal transformers often do not work out of the box for downstream multimodal tasks due to unseen joint distributions from the two modalities. To bridge this gap, Nagrani et al. (2021) introduced a few bottleneck (prompt) tokens to the self-attention mechanism, which are appended to the concatenation sequence of the two modalities to perform cross-modality feature interaction. Such PEFT mechanism is referred to as the *interactive prompting* paradigm in literature (Li et al., 2023).

Let us denote the collection of $B$ prompt tokens as $\mathbf{z}_{\text{fsn}} = [z_1, z_2, \ldots, z_B]_{\text{fsn}}$ to fuse two pre-trained transformers for two modalities (the subscript fsn is the shorthand for "fusion"). Starting from the pre-defined $L_f$-th layer (the fusion layer), the output tokens at the $(l+1)$-th layer for $l \in \{L_f, L_f + 1, \ldots, L - 1\}$ are computed as

$$
\begin{aligned}
\begin{bmatrix} \mathbf{z}_m^{l+1} \\ \hat{\mathbf{z}}_{\text{fsn}_m}^{l+1} \end{bmatrix} &= \mathbf{TransformerLayer}\left(\begin{bmatrix} \mathbf{z}_m^l \\ \mathbf{z}_{\text{fsn}}^l \end{bmatrix}\right), \\
\mathbf{z}_{\text{fsn}}^{l+1} &= \frac{1}{N_m} \sum_{m=1}^{N_m} \hat{\mathbf{z}}_{\text{fsn}_m}^{l+1},
\end{aligned}
\tag{3}
$$

where only the initial prompt token $\mathbf{z}_{\text{fsn}}^{L_f} := \mathbf{z}_{\text{fsn}}$ is tunable and shared, $m$ indexes a certain modality, and $N_m$ is the number of total modalities, which equates to 2 in vision–language fusion. We note the interaction among modalities is reflected by mixing modality-specific tokens $\hat{\mathbf{z}}_{\text{fsn}_m}^{l+1}$'s into the bottleneck token $\mathbf{z}_{\text{fsn}}^{l+1}$, and we will shortly revisit the mechanism and its limitations in Section 4.1.

### 3.3 Low-rank adaptation

Low-rank adaptation (LoRA; Hu et al. 2022) techniques have been widely adopted for parameter-efficient fine-tuning of large pre-trained models and serve as a more flexible alternative to prompting. We will discuss the incorporation of LoRA to strengthen the representation power in Section 4.3.

Given a pre-trained weight matrix $\mathbf{W}_0 \in \mathbb{R}^{d \times d'}$, LoRA constrains its update in a low-rank form: $\mathbf{W}_0 + \Delta\mathbf{W} = \mathbf{W}_0 + \mathbf{B}\mathbf{A}$, where $\mathbf{B} \in \mathbb{R}^{d \times r}$, $\mathbf{A} \in \mathbb{R}^{r \times d'}$, and the rank $r \ll \min(d, d')$. LoRA trains the matrices $\mathbf{A}$ and $\mathbf{B}$ instead of the large weight matrix $\mathbf{W}_0$, significantly lowering the memory requirements for tuning pre-trained models.

## 4 Enhancing Multimodal Fusion with Representation-centric Prompting

In vision–language fusion, the backbone architectures are usually transformer-based, and notably, the concept "prompt" is closely related to the attention mechanism. It is thus meaningful to take the characteristics of attention into consideration when enhancing the representation in interactive prompting. We dedicate

this section to exploring how prompts with information from another modality will impact the attention output and how we can accordingly incorporate attention-specific techniques into the interactive prompting paradigm.

### 4.1 Limitation on representation steering

We first present the limitation of existing interactive prompting paradigms, as a motivation for our incorporation of representation learning techniques.

Firstly, the literature on PEFT in NLP suggests that solely applying prompt tuning is usually inferior to adapter-tuning or LoRA (He et al., 2022; Chen et al., 2022b). They reported such prompts have limited impacts on the value part of attention, as well as existing attention scores among query tokens and key tokens. As prompting is deeply rooted in the unimodal studies on NLP, we note that similar limitations apply to vision–language fusion as well.

To further illustrate it, we derive the attention output in layer $L_f + 1$ (where the information from another modality has already been incorporated into the prompt tokens $\mathbf{z}_{\text{fsn}}^{L_f+1}$, as shown in Equation (3)). (From then on, we omit the layer index $L_f + 1$ for notational clarity). In that layer, the concatenated sequence of input tokens for modality $m$ is $\tilde{\mathbf{z}}_m = \begin{bmatrix} \mathbf{z}_m \\ \mathbf{z}_{\text{fsn}} \end{bmatrix}$, which includes both the original modality tokens $\mathbf{z}_m$ and the interactive prompt tokens $\mathbf{z}_{\text{fsn}}$.

We then locate the impact of prompts on attention outputs for a specific token $\mathbf{z}_{m,i}$ (the $i$-th row in the original sequence $\mathbf{z}_m$). The corresponding representation of its attention output is

$$\text{Attn}(\tilde{\mathbf{Q}}_i, \tilde{\mathbf{K}}, \tilde{\mathbf{V}}) = \text{softmax}\left(\tilde{\mathbf{Q}}_i \tilde{\mathbf{K}}^\top / \sqrt{d}\right) \tilde{\mathbf{V}},$$

where $\{\tilde{\mathbf{Q}}, \tilde{\mathbf{K}}, \tilde{\mathbf{V}}\} = \tilde{\mathbf{z}}_m \mathbf{W}_{\{Q,K,V\}}$ and $\tilde{\mathbf{Q}}_i = \tilde{\mathbf{z}}_{m,i} \mathbf{W}_Q$. In computing the attention scores, we separate $\tilde{\mathbf{K}}$ as $\mathbf{K}_m, \mathbf{K}_{\text{fsn}} = \mathbf{z}_m \mathbf{W}_K, \mathbf{z}_{\text{fsn}} \mathbf{W}_K$ and individually calculate their "similarity" / attention score with $\tilde{\mathbf{Q}}_i$ as:

$$[\mathbf{S}_{m,i}^{(m)}, \mathbf{S}_{m,i}^{(\text{fsn})}] := \text{softmax}\left(\frac{[\tilde{\mathbf{Q}}_i \mathbf{K}_m^\top, \tilde{\mathbf{Q}}_i \mathbf{K}_{\text{fsn}}^\top]}{\sqrt{d}}\right).$$

We note $\mathbf{z}_{\text{fsn}}$ **in this layer does not affect** $\tilde{\mathbf{Q}}_i \mathbf{K}_m^\top$, and the softmax operator will only reduce the corresponding attention scores $\mathbf{S}_{m,i}^{(m)}$ by a small factor. The consequent attention output for token $\mathbf{z}_{m,i}$ will be

$$\mathbf{S}_{m,i}^{(m)} \mathbf{V}_m + \mathbf{S}_{m,i}^{(\text{fsn})} \mathbf{V}_{\text{fsn}}, \tag{4}$$

in which $\mathbf{V}_m$ is formally invariant to the prompt $\mathbf{z}_{\text{fsn}}$ as well.

In addition to the conceptual limitation above, existing prompting-based vision–language fusion designs may further restrict representation steering in practice. First, the prompt channel itself is often narrow. For example, the **number of prompt tokens** ($B = 4$ in Nagrani et al. 2021) are much smaller than the counterparts in natural language processing tasks, which further limits the impact of $\mathbf{z}_{\text{fsn}}$ in Equation (4). Recent prompting-based fusion methods such as MoPE (Jiang et al., 2024) alleviate this issue by improving the adaptivity and expressiveness of prompts through a Mixture-of-Experts mechanism. Specifically, MoPE makes the prompt tokens $\mathbf{z}_{\text{fsn}}$ instance-dependent and thus strengthens the prompt term $\mathbf{S}_{m,i}^{(\text{fsn})} \mathbf{V}_{\text{fsn}}$. However, it still operates within the same prompting framework: the original value matrix $\mathbf{V}_m = \mathbf{z}_m \mathbf{W}_V$ remains unchanged, and the intra-modal similarity term $\tilde{\mathbf{Q}}_i \mathbf{K}_m^\top$ among original tokens is not directly modified. Therefore, MoPE improves prompt quality, but does not remove the structural limitation discussed above.

Second, for parameter efficiency, prompting-based fusion is often inserted only near the top of the transformer (e.g., PMF; Li et al. 2023). As a result, Equation (4) is already close to the final attention output, leaving limited room for subsequent layers to further propagate cross-modal influence into the original modality tokens. In this regard, prompt-driven representations alone may still be insufficient for steering the final transformer outputs.

Realizing these representation limitations, we are motivated to incorporate the representation learning techniques into prompting-based vision–language fusion, to **enhance the representation power of transformer outputs**. This direct idea provides a new channel for tackling this multimodal task, in addition to enhancing prompt tokens. We revisit the core components in the attention mechanism in Section 4.2 and propose our LoRA-assisted resolutions, which utilize the special characteristics of attention, in Section 4.3 and Section 4.4, respectively.

## 4.2 Locating the critical parameter subset

In implementing the direct idea, we note training the entire unimodal encoders is computationally demanding and contradicts the essence of prompt-tuning approaches. We are thus inspired to decompose the attention modules within transformers in the hope that **training a smaller subset of parameters will suffice**. Our following analyses start with rewriting the attention modules $\text{Attn}(\cdot)$ in a nonparametric estimator form (c.f. Chen et al. 2022a;b; more in-depth and self-contained derivations are deferred to Appendix C.2).

**Decomposition of attention in prompting.** Rather than introducing auxiliary prompting architecture as done in PMF, we take a targeted approach by revisiting the internals of the attention operation and identifying the key subcomponents that most directly influence the representation output. As suggested by Chen et al. (2022a;b), the representation learning in attention can be divided into parts ① the query–key interactions (i.e., the gram matrix $\mathbf{Q}\mathbf{K}^\top$) and ② the value $\mathbf{V}_m$. Following this framework, we similarly derive the case for attention bottleneck fusion in the next several paragraphs.

With the attention bottleneck formulation denoted above (c.f. Section 3.2), we recall the input $\tilde{\mathbf{z}}_m$ defined in Section 4.1, the query, key, and value matrices $\{\tilde{\mathbf{Q}}, \tilde{\mathbf{K}}, \tilde{\mathbf{V}}\} = \tilde{\mathbf{z}}_m \mathbf{W}_{\{Q,K,V\}}$ for the complete sequence $\tilde{\mathbf{z}}_m$, and the attention operation

$$\text{Attn}(\tilde{\mathbf{Q}}, \tilde{\mathbf{K}}, \tilde{\mathbf{V}}) = \text{softmax}\left(\frac{\tilde{\mathbf{Q}}\tilde{\mathbf{K}}^\top}{\sqrt{d}}\right)\tilde{\mathbf{V}}.$$

The first part of attention involves the similarity matrix $\tilde{\mathbf{S}}$ (corresponding to ① the query–key interactions), which can be decomposed separately for the query $\mathbf{Q}_m = \mathbf{z}_m \mathbf{W}_Q$ and the bottleneck counterpart $\mathbf{Q}_{\text{fsn}} = \mathbf{z}_{\text{fsn}} \mathbf{W}_Q$ as

$$\begin{aligned}
\tilde{\mathbf{S}} &= \text{softmax}\left(\frac{\tilde{\mathbf{Q}}\tilde{\mathbf{K}}^\top}{\sqrt{d}}\right) \\
&= \text{softmax}\left(\frac{1}{\sqrt{d}}\begin{bmatrix}\mathbf{Q}_m \tilde{\mathbf{K}}^\top \\ \mathbf{Q}_{\text{fsn}} \hat{\mathbf{K}}^\top\end{bmatrix}\right) = \begin{bmatrix}\mathbf{S}_m \\ \mathbf{S}_{\text{fsn}}\end{bmatrix}.
\end{aligned} \tag{5}$$

Notably due to the softmax operation, all values within the similarity matrix $\tilde{S}$ are bounded in $[0, 1]$.

Putting together ① the query–key interactions above and ② the value $\mathbf{V}_m$, we have

$$\text{Attn}\left(\tilde{\mathbf{Q}} = \begin{bmatrix}\mathbf{Q}_m \\ \mathbf{Q}_{\text{fsn}}\end{bmatrix}, \tilde{\mathbf{K}}, \tilde{\mathbf{V}}\right) = \begin{bmatrix}\mathbf{S}_m \tilde{\mathbf{V}} \\ \mathbf{S}_{\text{fsn}} \tilde{\mathbf{V}}\end{bmatrix}. \tag{6}$$

As shown in the preceding display, the output of the attention operation in attention bottleneck fusion consists of two parts, differentiated by distinct attention score matrices. The first corresponds to computing the attention score between the modality query $\mathbf{Q}_m$ and the concatenated key $\tilde{\mathbf{K}}$; the other instead computes the attention score for the bottleneck query $\mathbf{Q}_{\text{fsn}}$. Both are then multiplied with the joint value matrix $\tilde{\mathbf{V}}$.

We draw multiple crucial observations from the decomposition above. **Firstly**, the component $\mathbf{S}_m \tilde{\mathbf{V}}$ in Equation (6) is generally considered as more important than the other. $\mathbf{S}_m \tilde{\mathbf{V}}$ corresponds to the modality tokens $\mathbf{z}_m$ of this transformer layer (c.f. Equation (3)) and has a direct impact on the final prediction. **Moreover**, for $\mathbf{S}_m \tilde{\mathbf{V}}$ the query weight matrix ($\mathbf{W}_Q$) is only applied to the modality tokens $\mathbf{z}_m$, whereas other weight matrices ($\mathbf{W}_{\{K,V\}}$) are jointly applied to $\mathbf{z}_m$ and $\mathbf{z}_{\text{fsn}}$. As such, it is reasonable to speculate that the impact of training $\mathbf{W}_K$ and $\mathbf{W}_V$ is thus generally more significant than $\mathbf{W}_Q$. **Lastly**, we highlight that the attention score matrix $\tilde{S}$ contains bounded values in $[0, 1]$, whereas the values in $\tilde{\mathbf{V}}$ are unbounded. This intuitive observation implies that the value matrix $\tilde{\mathbf{V}}$ contributes more to the final output than the key matrix $\hat{\mathbf{K}}$, suggesting an emphasis on training $\mathbf{W}_V$ and aligning with Chen et al. (2022a); He et al. (2022).

Table 1: A motivating toy experiment on MM-IMDB examining the effect of adapting different attention components in the last transformer layer. Under a similar number of trainable parameters $|\theta|$, adapting the value projection ($\mathbf{W}_V^{[-1]}$; the superscript $[-1]$ denotes the last layer) achieves the largest improvement compared to adapting $\mathbf{W}_Q^{[-1]}$ or $\mathbf{W}_K^{[-1]}$. This suggests that the value is particularly important for improving multimodal representations. Results are reported as mean $\pm$ standard deviation over 5 runs.

| $\theta$ | $|\theta|$ | F1-Micro/Macro (%) |
|---|---|---|
| $\{\mathbf{z}_{\text{fsn}}\}$ | $<$0.1M ($<$0.05%) | 64.65±0.27/54.82±0.26 |
| $\{\mathbf{z}_{\text{fsn}}, \mathbf{W}_Q^{[-1]}\}$ | 1.2M (0.61%) | 65.48±0.23/55.79±0.21 |
| $\{\mathbf{z}_{\text{fsn}}, \mathbf{W}_K^{[-1]}\}$ | 1.2M (0.61%) | 65.63±0.11/56.21±0.17 |
| $\{\mathbf{z}_{\text{fsn}}, \mathbf{W}_V^{[-1]}\}$ | 1.2M (0.61%) | **66.24**±0.24/**58.57**±0.28 |

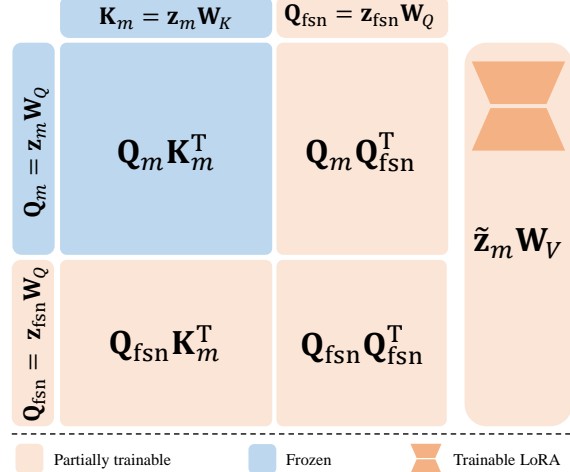

Figure 2: Illustration of the proposed representation-centric scheme within attention modules. The value part is adapted via LoRA on the value projection ($\mathbf{W}_V$), while in the bottom right corner of the attention score matrix the key matrix corresponding to bottleneck tokens (i.e., the original $\mathbf{K}_{\text{fsn}}$) is replaced by a query counterpart ($\mathbf{Q}_{\text{fsn}}$ at the top right corner). Scaling and softmax normalization are omitted here for clarity.

**Empirical validation.** From the decomposition (6) and the associated observations above, we obtain an intuition for multimodal fusion that $\mathbf{W}_V$ can as well be **the crucial subset of parameters** in a transformer layer. We perform sanity-check experiments to verify the intuition above in Table 1. Specifically, we begin with the attention bottleneck baseline (c.f. Nagrani et al. 2021; described in Section 3.2), then proceed to respectively train $\mathbf{W}_{\{Q,K,V\}}$ in the final transformer layer of BERT (Devlin et al., 2019) and ViT (Dosovitskiy, 2020).

We measure performance across all configurations above over 5 runs, reporting the average and associated standard deviation of the two metrics on a common benchmark MM-IMDB (Arevalo et al., 2017). Although incorporating those weight matrices requires the same number of trainable parameters, training $\mathbf{W}_V$ notably outperforms the others by a significant margin, with training $\mathbf{W}_K$ ranking second-best. This empirical validation reinforces our earlier intuitions, emphasizing the importance of training $\mathbf{W}_V$ in attention modules.

### 4.3 Value-based multimodal adaptation

With our analyses and observations above, upon attention bottleneck fusion we devise a representation-centric adaptation strategy to enhance its cross-modality representation power. Particularly in this subsection, we incorporate the LoRA (Hu et al., 2022) technique into attention bottleneck fusion.

Inspired by the attention module decomposition (6) and our toy experiments in Table 1, the first part of our proposed method is to exclusively tune the value weight matrix $\mathbf{W}_V$ while freezing the other weight

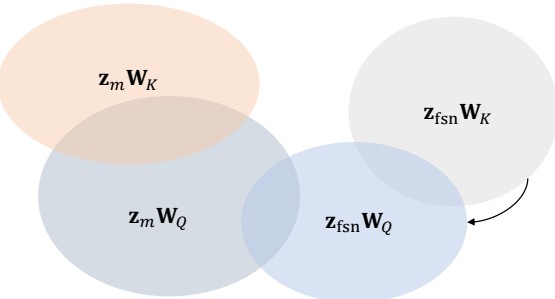

Figure 3: Intuition behind the $KQ$ replacement. For the modality token $\mathbf{z}_m$, its attention output naturally concerns the intra-modal product, $\mathbf{z}_m\mathbf{W}_Q(\mathbf{z}_m\mathbf{W}_K)^\top$, and the modality–bottleneck product, $\mathbf{z}_m\mathbf{W}_Q(\mathbf{z}_{\text{fsn}}\mathbf{W}_K)^\top$. The former is naturally aligned in pre-training, while the latter can be less well matched since the bottleneck token $\mathbf{z}_{\text{fsn}}$ are introduced afterwards during fusion. The $KQ$ replacement reduces this mismatch by projecting bottleneck tokens as well with $\mathbf{W}_Q$ rather than $\mathbf{W}_K$, i.e., replacing $\mathbf{z}_{\text{fsn}}\mathbf{W}_K$ with $\mathbf{z}_{\text{fsn}}\mathbf{W}_Q$ to facilitate alignment with $\mathbf{z}_m\mathbf{W}_Q$.

matrices, as shown in Figure 2. For the pre-trained value weight matrix $\mathbf{W}_V$, we can further reduce the size of trainable parameters by applying LoRA to tune $\mathbf{W}_V$. We note that our main focus in this paper is to demonstrate how representation-centric analysis can guide the design in multimodal fusion; we do not expect the LoRA module design to be optimal, and it is possible that simply adopting another advanced LoRA approach can further improve.

Formally, for the $l$-th transformer layer defined in Equation (2), the attention operation $\text{MHA}^l(\mathbf{x})$ with LoRA on $\mathbf{W}_V$ is then computed as

$$\text{Attn}\left(\mathbf{x}\mathbf{W}_Q^l, \mathbf{x}\mathbf{W}_K^l, \mathbf{x}\left(\mathbf{W}_V^l + \frac{\alpha^l}{r^l}\mathbf{B}^l\mathbf{A}^l\right)\right),$$

with pre-trained (fixed) weight matrices $\mathbf{W}_{\{Q,K,V\}}^l$ and trainable parameters $\mathbf{B}^l$ and $\mathbf{A}^l$. Here, the constants $\alpha^l$ and $r^l$ are tunable scalar hyperparameters; $r^l$ represents the LoRA rank, and $\alpha^l = 2r^l$.

### 4.4 Key–query replacement for bridging cross-modal gaps in bottleneck

While the pre-trained weight matrices work out of the box for the modality tokens $\mathbf{z}_m$, they may not be ideal for prompt tokens $\mathbf{z}_{\text{fsn}}$ (which never appear in pre-training). To have a clearer picture of this, we rewrite the similarity matrix as

$$\tilde{\mathbf{S}} = \text{softmax}\left(\frac{1}{\sqrt{d}}\begin{bmatrix}\mathbf{Q}_m\tilde{\mathbf{K}}^\top \\ \mathbf{Q}_{\text{fsn}}\tilde{\mathbf{K}}^\top\end{bmatrix}\right)$$
$$= \text{softmax}\left(\frac{1}{\sqrt{d}}\begin{bmatrix}\mathbf{Q}_m\mathbf{K}_m^\top & \mathbf{Q}_m\mathbf{K}_{\text{fsn}}^\top \\ \mathbf{Q}_{\text{fsn}}\mathbf{K}_m^\top & \mathbf{Q}_{\text{fsn}}\mathbf{K}_{\text{fsn}}^\top\end{bmatrix}\right),$$

where we recall $\{\mathbf{Q},\mathbf{K}\}_m = \mathbf{z}_m\mathbf{W}_{\{Q,K\}}$ and $\{\mathbf{Q},\mathbf{K}\}_{\text{fsn}} = \mathbf{z}_{\text{fsn}}\mathbf{W}_{\{Q,K\}}$. In the first part of $\tilde{\mathbf{S}}$ that corresponds to the modality token, the original gram matrix $\mathbf{Q}_m\mathbf{K}_m^\top$ would work immediately thanks to its appearance in pre-training, while the remaining part $\mathbf{Q}_m\mathbf{K}_{\text{fsn}}^\top$ may not align well due to the distinction in distributions of the modality tokens and the prompt tokens. (The second part of $\tilde{\mathbf{S}}$, namely $\mathbf{Q}_{\text{fsn}}\tilde{\mathbf{K}}^\top$, does not raise similar concerns as the prompt tokens themselves are tunable.) Moreover, our first technique proposes to freeze the query and key weight matrices $\mathbf{W}_{\{Q,K\}}$, which deteriorates the alignment problem during optimization.

To address this potential issue, we further propose to replace the key matrix corresponding to the prompt tokens ($\mathbf{K}_{\text{fsn}}$), with the query counterpart ($\mathbf{Q}_{\text{fsn}}$). As both $\mathbf{Q}_m$ and $\mathbf{Q}_{\text{fsn}}$ utilize the same projection matrix $\mathbf{W}_Q$, this intuitively harmonizes the distribution discrepancy in a new modality–bottleneck product $\mathbf{Q}_m\mathbf{Q}_{\text{fsn}}^\top$. Specifically, in obtaining the attention score matrix defined in Equation (5), instead of computing $\tilde{\mathbf{K}} =$

$\tilde{\mathbf{z}}_m \mathbf{W}_K$ as a whole, we only apply $\mathbf{W}_K$ to $\mathbf{z}_m$ while project $\mathbf{z}_{\text{fsn}}$ with the query weight matrix $\mathbf{W}_Q$. This leads to the following attention score matrix

$$\tilde{\mathbf{S}} = \text{softmax}\left(\frac{1}{\sqrt{d}} \begin{bmatrix} \mathbf{Q}_m \mathbf{K}_m^\top & \mathbf{Q}_m \mathbf{Q}_{\text{fsn}}^\top \\ \mathbf{Q}_{\text{fsn}} \mathbf{K}_m^\top & \mathbf{Q}_{\text{fsn}} \mathbf{Q}_{\text{fsn}}^\top \end{bmatrix}\right).$$

The approach is illustrated in Figure 2, and the intuition within is visualized in Figure 3. We further provide modality-to-prompt attention score (the upper-right corner) visualizations in Appendix A.5 to examine the alignment directly affected by this replacement. We ablate this proposal in Section 5.3.

## 5 Empirical Results

Following the protocols in previous works (Li et al., 2023), we test the performance of our methods, induced from the representation-centric perspective, on vision–language fusion tasks as follows.

### 5.1 Benchmarking datasets and baselines

**Benchmarking datasets.** We conduct experiments on Multibench (Liang et al., 2021), including (1) multi-label movie genre classification on **MM-IMDB** (Arevalo et al., 2017), which consists of movie plot outline and poster pairs; (2) multimodal classification on **UPMC Food-101** (Wang et al., 2015), which contains food images with recipe descriptions for 101 types of food; (3) visual entailments on **SNLI-VE** (Xie et al., 2019), which aim to reason the relationship between an image premise and a text hypothesis, categorized into entailment, contradiction, and neutrality. We report Macro and Micro F1-scores for MM-IMDB, and classification accuracy for UPMC Food-101 and SNLI-VE. Due to space limit, more details on these benchmarks are deferred to Appendix A.1.

**Baseline methods.** We categorize baseline methods into two branches: ❶ full fine-tuning, where unimodal encoders are jointly trained with fusion modules and classification heads; ❷ prompt-tuning, where unimodal encoders are kept fixed during training, and only a small set of parameters (prompt tokens) is tuned. The latter is often preferred because of its significantly lower computational requirement, whereas full fine-tuning generally achieves higher performance due to its stronger modeling capacity.

Among full fine-tuning baselines, we include single-modality fine-tuning with ViT (Dosovitskiy, 2020) and BERT (Devlin et al., 2019). Other fine-tuning-based multimodal fusion methods are reported as well, including late fusion with concatenation (LateConcat) results from Li et al. (2023), multimodal fusion with attention bottleneck (MBT) (Nagrani et al., 2021), and MMBT (Kiela et al., 2019).

We mainly compare with multiple existing prompt-tuning approaches. The baselines include PromptFuse and BlindPrompt proposed in Liang et al. (2022), prompting-based late concatenation (P-LateConcat, whose results are reported in Li et al. (2023)), prompting-based attention bottleneck (P-MBT, an extension of MBT where pre-trained encoders are kept frozen), and PMF (Li et al., 2023), which additionally introduces cross-modal projection between prompt tokens. Moreover, we further include MoPE (Jiang et al., 2024), a recent strong prompt-tuning baseline that integrates a Mixture-of-Experts (MoE) design into prompt tuning to generate suitable prompts.

The results for the compared methods on MM-IMDB, Food-101, and SNLI-VE are primarily taken from Li et al. (2023). In particular, the performance of all fine-tuning methods are taken from Li et al. (2023). For prompt-tuning methods, the results, except for P-MBT, MoPE and ours, are as well taken from Li et al. (2023). (Our reproduction of P-MBT attains higher performance than reported in Li et al. (2023).) Particularly, the results of MoPE (Jiang et al., 2024) are obtained from our reproduction, considering that the original MoPE adopts Swin Transformer (Liu et al., 2021), rather than ViT (c.f. **??** for more detail).

Table 2: Experimental results on MM-IMDB, UPMC Food-101, and SNLI-VE. Full fine-tuning baselines are highlighted in gray and are provided as references, while our main comparison focuses on prompt-tuning methods. We report the number of trainable parameters (# Trainable Params.), excluding classification heads for consistency across datasets. Following the experimental setting of PMF (Li et al., 2023), all results are reported as the mean over 3 runs with different random seeds. Compared with MoPE (Jiang et al., 2024), our improvements are statistically significant on most metrics under a 95% significance level (∗ denotes statistical significance); see Appendix A.3 for details. Values in parentheses (% total) indicate the proportion relative to the total number of parameters in the original model.

| | | # of Trainable Param. (% total) | MM-IMDB | | Food-101 | SNLI-VE |
| | | | F1-Micro (%) | F1-Macro (%) | Acc. (%) | Acc. (%) |
|---|---|---|---|---|---|---|
| *fine-tuning* | ViT | 86.5M (44.02%) | 49.88 | 38.39 | 74.69 | 33.33 |
| | BERT | 109.0M (55.47%) | 64.31 | 58.00 | 87.44 | 69.82 |
| | LateConcat | 196.0M (99.75%) | 64.92 | 59.56 | 93.29 | 70.01 |
| | MBT | 196.0M (99.75%) | 64.81 | 59.60 | 93.56 | 74.02 |
| | MMBT | 196.5M (100%) | 66.10 | 60.80 | 94.10 | 74.69 |
| *prompt-tuning* | PromptFuse | <0.1M (<0.05%) | 54.49 | 48.59 | 82.21 | 64.53 |
| | BlindPrompt | <0.1M (<0.05%) | 56.46 | 50.18 | 84.56 | 65.54 |
| | P-LateConcat | 0.3M (0.15%) | 59.93 | 53.91 | 89.03 | 63.05 |
| | P-MBT | <0.1M (<0.05%) | 64.86 | 56.45 | 89.88 | 62.79 |
| | PMF | 2.5M (1.27%) | 64.51 | 58.77 | 91.51 | 71.92 |
| | MoPE ($k = 4$, ViT) | 1.5M (0.76%) | 67.08 | 60.51 | 92.21 | 71.52 |
| | **Ours** | 0.20M (0.10%) | **68.09**∗ | **60.92** | **93.53**∗ | **72.31**∗ |

## 5.2 Main results and discussion

We report the results in Table 2. Using the same pre-trained unimodal transformers, our method outperforms all involved prompting-based fusion methods on all benchmarks. Notably, our method is superior to strong prompting-based baselines such as MoPE, while using fewer trainable parameters.

We recall that several previous prompting-based methods rely on more complex architectural designs. For example, PMF (Li et al., 2023) splits prompt tokens into three sets and introduces cross-modality projection, while MoPE (Jiang et al., 2024) further incorporates a Mixture-of-Experts (MoE) design and uses three prompt types (Static, Dynamic, and Mapped), whereas our approach remains comparatively simple. This suggests that improved multimodal fusion can be achieved without introducing additional prompt-engineering complexity.

We also compare our method with full fine-tuning methods, highlighted in gray in Table 2. Although prior prompting-based methods achieve strong results on the relatively small MM-IMDB dataset, they still lag behind full fine-tuning methods on the larger benchmarks considered here, namely Food-101 and SNLI-VE. Our proposed method further narrows the gap on these larger-scale datasets. These results suggest that our representation-centric prompting approach is more effective on the larger benchmarks considered here than previous prompt-tuning methods.

## 5.3 Ablation studies

To assess the contribution of each proposed component, we conduct ablation studies following the same experimental setup on all three datasets as above. In particular, we aim to answer three questions: (1) whether direct representation adaptation is more effective than prompt-only designs, (2) whether the gains of our method primarily come from $V$ LoRA and are further improved by the $KQ$ replacement, and (3) whether the advantage of representation-centric fusion persists with larger backbones. Due to space limit, The complexity analysis and empirical runtime comparison are provided in Appendix A.6. Appendix A.7 further investigates the usage of larger LoRA ranks and shows that increasing the trainable parameter count does not necessarily improve performance.

**Representation adaptation via LoRA already provides strong gains, and combining it with MBT yields the best performance.** To isolate the effect of representation adaptation, we report a set of late-fusion PEFT variants that adapt the unimodal encoders with LoRA or adapters and fuse the final

Table 3: Ablation results isolating the effect of representation adaptation across three benchmarks. We compare several late-fusion PEFT variants, where PEFT modules are inserted into the unimodal encoders and image/text representations are fused only at the classifier level, without explicit cross-modal fusion before the classifier. These variants already improve over the prompting baseline PMF, indicating the efficacy of direct representation adaptation. Combining $V$ LoRA with the P-MBT framework and $KQ$ replacement ("Ours") further improves performance and achieves the best results. The improvement from the best late-fusion PEFT variant to "Ours" is statistically significant (under a 95% significance level; $*$ denotes significance). (Li et al. (2023) solely provided the average performance of PMF, which renders statistical comparison infeasible.)

| | $\|\theta\|$ (% total) | MM-IMDB F1-Micro/Macro (%) | Food-101 Acc. (%) | SNLI-VE Acc. (%) |
|---|---|---|---|---|
| PMF | 2.5M (1.27%) | 64.51/58.77 | 91.51 | 71.92 |
| *Late-fusion PEFT variants* | | | | |
| $Q$ LoRA | 0.20M (0.10%) | 65.77/57.02 | 92.79 | 68.26 |
| $K$ LoRA | 0.20M (0.10%) | 66.00/57.80 | 92.81 | 68.49 |
| $O$ LoRA | 0.20M (0.10%) | 66.81/58.48 | 92.88 | 68.85 |
| $V$ LoRA | 0.20M (0.10%) | 66.98/59.14 | 93.10 | 69.04 |
| $Q,K,V,O$ LoRA | 0.20M (0.10%) | 66.47/58.73 | 92.87 | 69.09 |
| $FFN$ LoRA | 0.25M (0.13%) | 64.68/56.23 | 92.77 | 69.20 |
| Adapter | 0.20M (0.10%) | 65.49/58.04 | 92.98 | 69.30 |
| **Ours** | 0.20M (0.10%) | **68.09**$^*$/**60.92**$^*$ | **93.53**$^*$ | **72.31**$^*$ |

Table 4: Ablation results for the *main components* of our method across MM-IMDB, Food-101, and SNLI-VE. Starting from the P-MBT baseline, we add $V$ LoRA and then the parameter-free $KQ$ replacement trick ($KQ$ Repl.); we also include $K$ LoRA and $V,K$ LoRA to compare against directly adapting the key projection. Implementation details are provided in Appendix A.4. Here, $*$ indicates 95% significance for $V$ LoRA over P-MBT, as well as "Ours" over $V$ LoRA.

| | $\|\theta\|$ (% total) | MM-IMDB F1-Micro/Macro (%) | Food-101 Acc. (%) | SNLI-VE Acc. (%) |
|---|---|---|---|---|
| P-MBT | <0.1M (<0.05%) | 64.86/56.45 | 89.88 | 62.79 |
| + $V$ LoRA | 0.20M (0.10%) | 67.34$^*$/59.32$^*$ | 93.13$^*$ | 71.52$^*$ |
| + $K$ LoRA | 0.20M (0.10%) | 66.31/57.77 | 92.84 | 70.58 |
| + $V,K$ LoRA | 0.25M (0.13%) | 67.53/60.29 | 93.29 | 71.64 |
| + $V$ LoRA, $KQ$ Repl. | 0.20M (0.10%) | **68.09**$^*$/**60.92**$^*$ | **93.53**$^*$ | **72.31**$^*$ |

image/text representations only at the classifier level (Table 3). The implementation details can be found in Appendix A.4. These late-fusion PEFT baselines are used to separate generic PEFT gains from the contribution of our full fusion design. On MM-IMDB, all single-projection LoRA variants improve Micro-F1 over PMF; in particular, $V$ LoRA improves Macro-F1, reaching 59.53% compared with 58.77% for PMF. On Food-101, all reported late-fusion PEFT variants outperform PMF, with accuracies between 92.77% and 93.10% compared with 91.51% for PMF. On SNLI-VE, the best reported PEFT baseline reaches 69.30%. It therefore suggests that a substantial part of the performance gain comes from directly adapting the internal representation, rather than from increasing prompt complexity alone.

Combining LoRA with the MBT-style bottleneck interaction ("Ours" in Table 3) further improves performance to 68.09%/60.92% on MM-IMDB, 93.53% on Food-101, and 72.31% on SNLI-VE. Across all three benchmarks, our method outperforms the late-fusion PEFT variants, indicating that direct encoder representation adaptation alone does not account for the full gain. This additional gain indicates that lightweight bottleneck interaction remains useful, but becomes most effective when built on top of stronger, adapted unimodal representations.

**$V$ LoRA provides a strong gain, and $KQ$ replacement brings a further improvement.** In Table 4, the attention-bottleneck baseline P-MBT attains 64.86%/56.45% in Micro/Macro F1 on MM-IMDB. Adding

Table 5: Ablation results on MM-IMDB with *larger backbones* (ViT-large + BERT-large). Our method continues to outperform the prompt-based baseline PMF-large, indicating that the benefit of representation-centric adaptation is not limited to the base-backbone setting.

| | $|\theta|$ (% total) | MM-IMDB F1-Micro/Macro (%) |
|---|---|---|
| PMF-large | 4.5M (2.29%) | 66.72/61.66 |
| Ours-large | 0.31M (0.16%) | **69.19/62.63** |

$V$ LoRA improves the result to 67.34%/59.32%, corresponding to gains of +2.48/+2.87 percentage points over P-MBT. This supports that adapting the value projection is an effective component, which is consistent with our earlier analysis that the value branch offers a more direct handle on the final attention output.

We next examine whether the improvement from $KQ$ replacement can be reproduced by directly learning the key projection. Recall that $KQ$ replacement modifies only the modality-to-prompt matching block: the original term $\mathbf{Q}_m\mathbf{K}_{\text{fsn}}^{\top}$ is replaced by $\mathbf{Q}_m\mathbf{Q}_{\text{fsn}}^{\top}$, while the modality-token keys and the value aggregation remain unchanged. A natural alternative is therefore to add LoRA to the key projection and let the model learn a better key representation. To test this alternative, Table 4 includes both $K$ LoRA and $V, K$ LoRA variants. The results show that key adaptation is useful but does not supersede key–query replacement. On MM-IMDB, $K$ LoRA improves P-MBT, but remains below the $V$ LoRA variant. Adding rank-2 $K$ LoRA on top of $V$ LoRA further improves the result to 67.53%/60.29%, but it uses more trainable parameters and remains below $V$ LoRA with key–query replacement, which reaches 68.09%/60.92%. The same comparison holds on Food-101 and SNLI-VE: $V, K$ LoRA reaches 93.29% and 71.64%, while $V$ LoRA with key–query replacement reaches 93.53% and 72.31%. The fact that this extra gain is obtained without increasing parameter count suggests that the improvement is not merely due to more trainable key capacity, but to the modified modality-to-prompt interactions induced by key–query replacement.

**The advantage persists with larger encoders.** On MM-IMDB with *ViT-large* and *BERT-large* backbones, our model attains 69.19%/62.63% in Micro/Macro F1, exceeding PMF-large (66.72%/61.66%) by +2.47/+0.97 percentage points, as shown in Table 5. This result is important for two reasons. First, it shows that the benefit of our method is not limited to the base-backbone regime or to relatively capacity-constrained models. Even when the unimodal encoders become substantially stronger, prompt-only fusion still leaves room for improvement, and representation-centric adaptation continues to provide gains. Second, the advantage is achieved with far fewer trainable parameters than PMF-large (0.31M vs. 4.5M), which suggests that the gain does not simply come from scaling adaptation size, but from placing adaptation on a more effective part of the fusion mechanism. Taken together, this ablation indicates that larger unimodal backbones do not eliminate the fusion bottleneck targeted by our method; instead, representation-centric adaptation remains complementary to backbone scaling.

## 6 Conclusion

In this work, we revisited prompting-based vision–language fusion from a representation-centric perspective. We demonstrated, both theoretically and empirically, that within frozen attention layers, prompt tokens have limited direct control over modality token representations. This motivates a simple alternative: directly adapting a small subset of attention components instead of further increasing prompt complexity. Specifically, we instantiate this idea with value-only low-rank adaptation and a lightweight key–query replacement strategy, yielding a parameter-efficient fusion design. Experiments show that the proposed method consistently outperforms prior prompting-based baselines while requiring fewer trainable parameters, and our ablations further confirm the effectiveness of representation-centric adaptation. Overall, these results suggest that in the frozen-encoder fusion setting studied here, allocating limited trainable parameters to attention components, as motivated by representation-centric analysis, can be more effective than refining prompt architectures alone. We further discuss how this representation-centric view may inform contemporary vision–language adaptation in Appendix B.

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

# A More on Empirical Results

## A.1 Dataset details

**MM-IMDB** is a large-scale, long-tailed multi-label classification benchmark for multimodal understanding. This dataset consists of 25,956 image-text pairs and each movie can be categorized into multiple genres. Each instance is composed of a movie poster image and its corresponding plot summary, with labels drawn from a set of 23 genre categories.

**UPMC Food-101** contains 90,840 image-text pairs, containing food images with textual recipe descriptions for 101 types of food. The sample size for each class ranges from 790 to 956. Originally, the dataset only contained training and testing sets, we thus follow previous works (Li et al., 2023; Kiela et al., 2019) to create the validation sets of 5000 samples from the training set.

**SNLI-VE** contains 565,286 image-text pairs, which aim to reason the relationship between an image premise and a text hypothesis, categorized into entailment, contradiction, and neutrality. We follow previous works (Li et al., 2023) to only use image premise and text hypothesis as inputs.

## A.2 Location of fusion layers and LoRA injection

In our experiments, we use ViT (Dosovitskiy, 2020) and BERT (Devlin et al., 2019) as unimodal encoders. For the *base* configuration, both backbones contain 12 Transformer layers (indexed from 0 to 11). We partition these layers into three regions using two hyperparameters: `n_fusion_layers` and `added_fusion_layers`. For the reported results, both are set to 4. Specifically, `n_fusion_layers=4` designates layers 8–11 as cross-modal fusion layers, where LoRA modules are also applied. In addition, `added_fusion_layers=4` extends LoRA injection to four earlier layers (layers 4–7), i.e., below the fusion region. For the *large* configuration solely in Table 5, both backbones contain 24 Transformer layers (indexed from 0 to 23). In the reported results, both `n_fusion_layers` and `added_fusion_layers` are set to 4 as well.

## A.3 Statistical significance testing

In Table 2 of the main text, we conduct paired $t$-tests to verify that the performance gains of our method over MoPE (Jiang et al., 2024) are statistically significant and not due to random variation.

For each dataset, strictly following the original experimental setting, both methods are trained under $N = 3$ independent runs with the same random seeds. We keep all other hyperparameters identical. Let $\{a_1, a_2, a_3\}$ and $\{b_1, b_2, b_3\}$ denote the per-run evaluation scores of our method and MoPE, respectively. We compute the paired differences $\delta_i = a_i - b_i$ and perform a one-sided paired $t$-test under the hypotheses

$$H_0 : \mu_\delta \leq 0 \quad \text{vs.} \quad H_1 : \mu_\delta > 0,$$

where $\mu_\delta$ is the population mean of the paired differences. The test statistic is

$$t = \frac{\bar{\delta}}{s_\delta/\sqrt{N}},$$

with $\bar{\delta}$ and $s_\delta$ denoting the sample mean and standard deviation of the differences, respectively. The $p$-value is obtained from the $t$-distribution with $N - 1 = 2$ degrees of freedom. We reject $H_0$ at significance level $\alpha = 0.05$.

Table 6 summarizes the results. Across all three datasets, the one-sided $p$-values are below the 0.05 threshold, and every 95% confidence interval for $\mu_\delta$ has a strictly positive lower bound, confirming that the improvements are statistically significant.

The 95% confidence intervals are computed as $\bar{\delta} \pm t_{\alpha/2,\,N-1} \cdot \mathrm{SE}$, where $t_{0.025,\,2} \approx 4.303$. Since our hypothesis is directional (our method > MoPE), the one-sided $p$-values are obtained by halving the two-tailed $p$-values reported by the paired $t$-test (i.e., 0.0197/2, 0.0017/2, and 0.0470/2 for SNLI-VE, Food-101, and MM-IMDB, respectively).

Table 6: Statistical significance test results (one-sided paired $t$-test, $N = 3$ runs, df $= 2$). $\bar{\delta}$: mean performance difference (ours $-$ MoPE); SE: standard error of the difference; CI: 95% confidence interval for $\mu_\delta$.

| Dataset (Metric) | $\bar{\delta}$ | SE | $t$ | $p$ (one-sided) | 95% CI |
|---|---|---|---|---|---|
| SNLI-VE (Acc) | 1.11% | 0.2% | 7.022 | 0.010 | [0.43%, 1.79%] |
| UPMC Food-101 (Acc) | 1.31% | 0.1% | 24.082 | <0.001 | [1.08%, 1.54%] |
| MM-IMDB (F1-micro) | 1.38% | 0.3% | 4.447 | 0.024 | [0.04%, 2.72%] |

Following the same paired t-test procedure, we conduct paired t-tests for the ablation studies in Tables 3 and 4. For Table 3, we test whether our method improves over the strongest late fusion PEFT baseline for each reported metric. For Table 4, we test two planned component comparisons: whether adding $V$ LoRA improves over P-MBT, and whether adding key–query replacement further improves over the $V$ LoRA variant.

## A.4  Implementation details

The experiment settings in previous works are strictly followed to ensure a fair comparison. We universally use the AdamW optimizer with a learning rate of $2 \times 10^{-4}$ for MM-IMDB, UPMC Food-101, and SNLI-VE. We set weight decay to $1 \times 10^{-2}$, number of prompt tokens to 4, and batch size to 32 for all experiments. Following prior work (Li et al., 2023), we decay the learning rate by a factor of 0.1 for 3 non-improving epochs and stop training for 7 non-improving epochs on all experiments. The experiment results are reported as the average over 3 runs, in accordance with Li et al. (2023). Each experiment on MM-IMDB, UPMC Food-101, and SNLI-VE takes approximately 2.3, 7.6, and 52.2 hours, respectively, on NVIDIA RTX 4090 GPUs, which varies across runs due to the different number of epochs required for convergence.

For our reproduction of MoPE (Jiang et al., 2024), we implement it under a ViT-based setting for a fair comparison, since the public GitHub repository only provides the Swin-Transformer version for the vision encoder. For the other setups, we follow the original MoPE setting and use the AdamW optimizer with a learning rate of $4 \times 10^{-4}$ for the main modality and $5 \times 10^{-4}$ for the complementary modality, using 4 experts ($k = 4$) in its MoE modules.

For the late-fusion PEFT variants reported in Table 3, we apply LoRA separately to each of the $Q$, $K$, $V$, and $O$ attention projection matrices with rank 8. We also include an all-attention LoRA variant, applying LoRA to all four attention projections with rank 2 to maintain a comparable trainable-parameter budget. For FFN-LoRA, we insert LoRA modules into the Transformer FFN layers, also with rank 2. For the Adapter variant, we use parameter-efficient adapters instead of LoRA modules. The adapter architecture follows Houlsby et al. (2019), and the adapters are inserted into the top eight layers. All these variants adapt unimodal encoders and fuse image/text representations only at the classifier level.

Regarding the implementation of our methods, by default, we follow the previous settings and use BERT-base and ViT-base as unimodal encoders, and LoRA rank as 8 unless specified otherwise. We build our method on the simple P-MBT baseline to highlight the effect of representation-centric adaptation and to avoid introducing additional prompt-engineering complexity.

## A.5  Attention heatmap under key–query replacement

To further analyze the effect of key–query replacement, we visualize the modality-to-prompt matching block at initialization in Figures 4 to 6, following the visualization practice of Lu et al. (2016). This visualization focuses on the exact attention block modified by key–query replacement, rather than the full self-attention matrix or trained attention probabilities. In the original design, modality tokens match bottleneck tokens through $\mathbf{Q}_m \mathbf{K}_{\text{fsn}}^\top$; with key–query replacement, this block becomes $\mathbf{Q}_m \mathbf{Q}_{\text{fsn}}^\top$, where the subscript $m$ denotes either image patch tokens or text tokens.

For image tokens, each heatmap cell corresponds to one ViT patch in the query matrix $\mathbf{Q}_m$. For text tokens, each heatmap cell corresponds to one text token. The color temperature denotes the pre-softmax matching

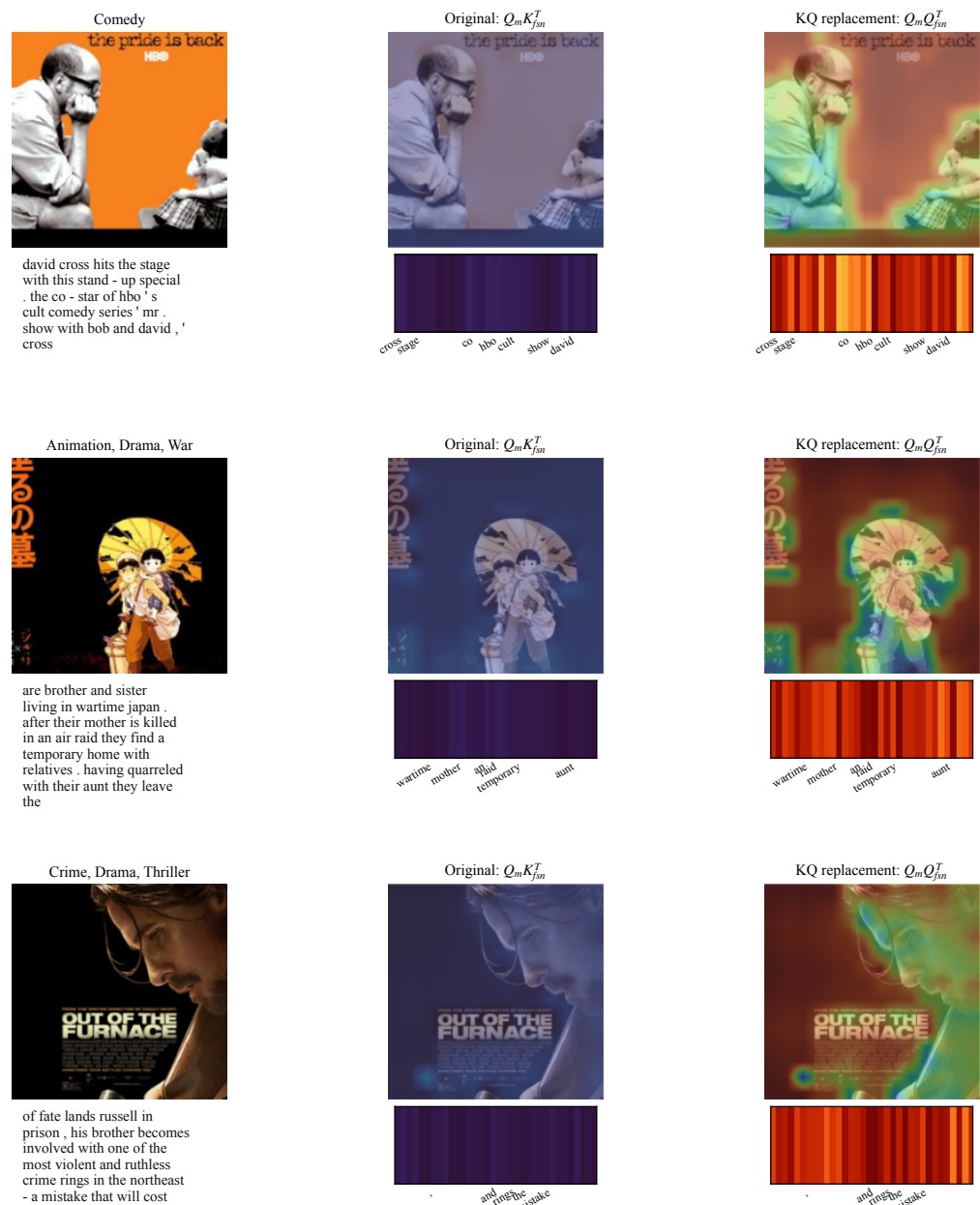

Figure 4: Attention heatmap: Visualization of modality-to-prompt interactions on MM-IMDB. The visualization shows the pre-softmax matching block between modality tokens and bottleneck prompt tokens, averaged over attention heads and prompt tokens.

score between the modality token and the prompt tokens, averaged over 12 attention heads and 4 prompt tokens. Therefore, these figures should not be interpreted as showing final semantic attention after training. Instead, they show how key–query replacement changes the initialization-stage matching between pretrained modality tokens and newly introduced bottleneck prompt tokens.

Across the three datasets, $\mathbf{Q}_m \mathbf{Q}_{\text{fsn}}^{\top}$ produces stronger modality-to-prompt interactions than $\mathbf{Q}_m \mathbf{K}_{\text{fsn}}^{\top}$ at initialization. This supports the motivation for key–query replacement: prompt tokens are newly initialized bottleneck tokens, so their original key vectors are not naturally matched to pretrained modality queries.

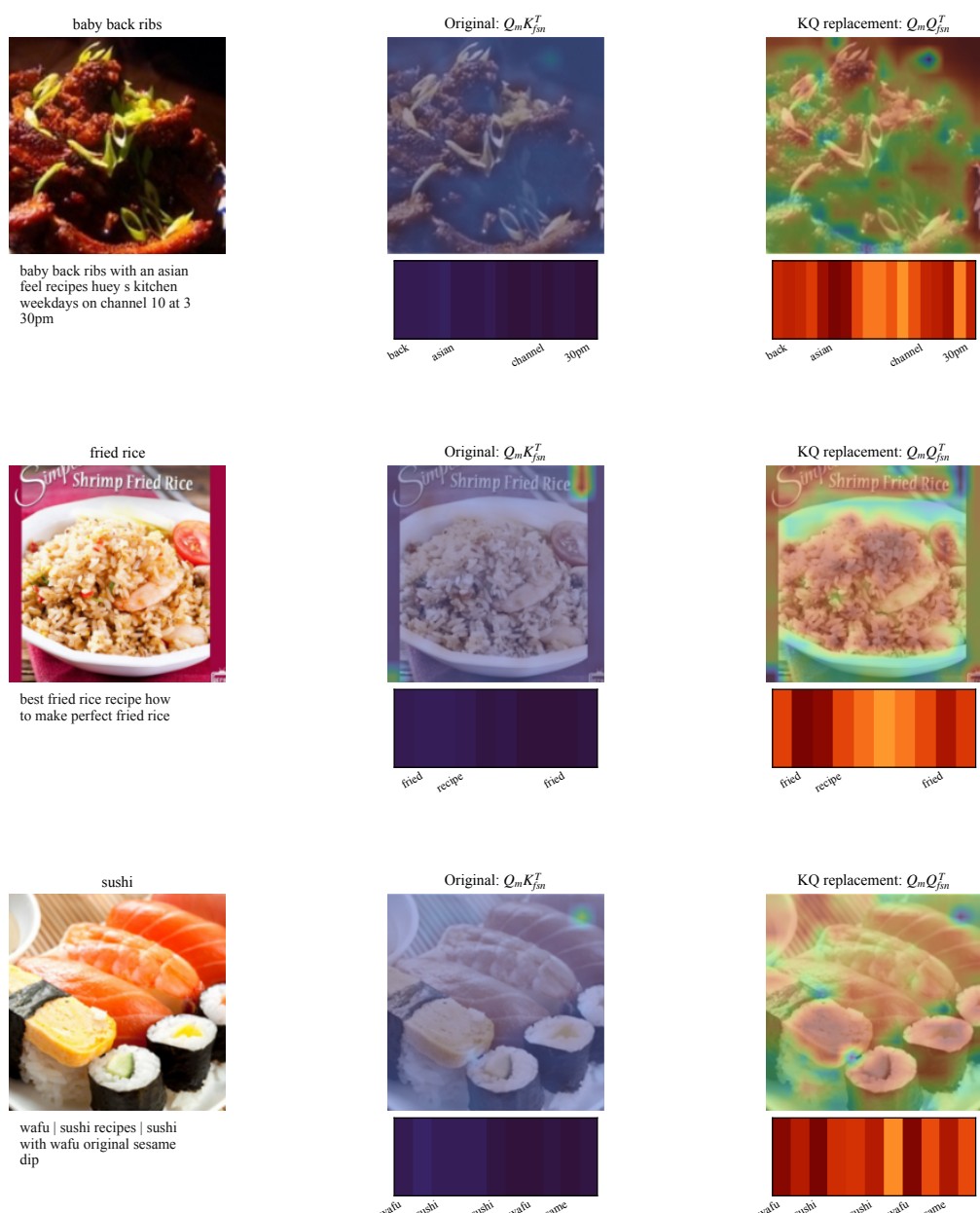

Figure 5: Attention heatmap: Visualization of modality-to-prompt interactions on Food-101. The original model uses $\mathbf{Q}_m \mathbf{K}_{\text{fsn}}^{\top}$, while key–query replacement uses $\mathbf{Q}_m \mathbf{Q}_{\text{fsn}}^{\top}$. key–query replacement yields stronger modality-to-prompt interactions prior to training.

Key–query replacement compares modality tokens and prompt tokens in the same query-projection space, making the bottleneck prompts more accessible to image and text tokens before training.

## A.6 Complexity and runtime

We compare the efficiency of our method with PMF from two perspectives: the parameter-level complexity of the fusion block and the measured inference latency. With unimodal encoders frozen, cost is dominated

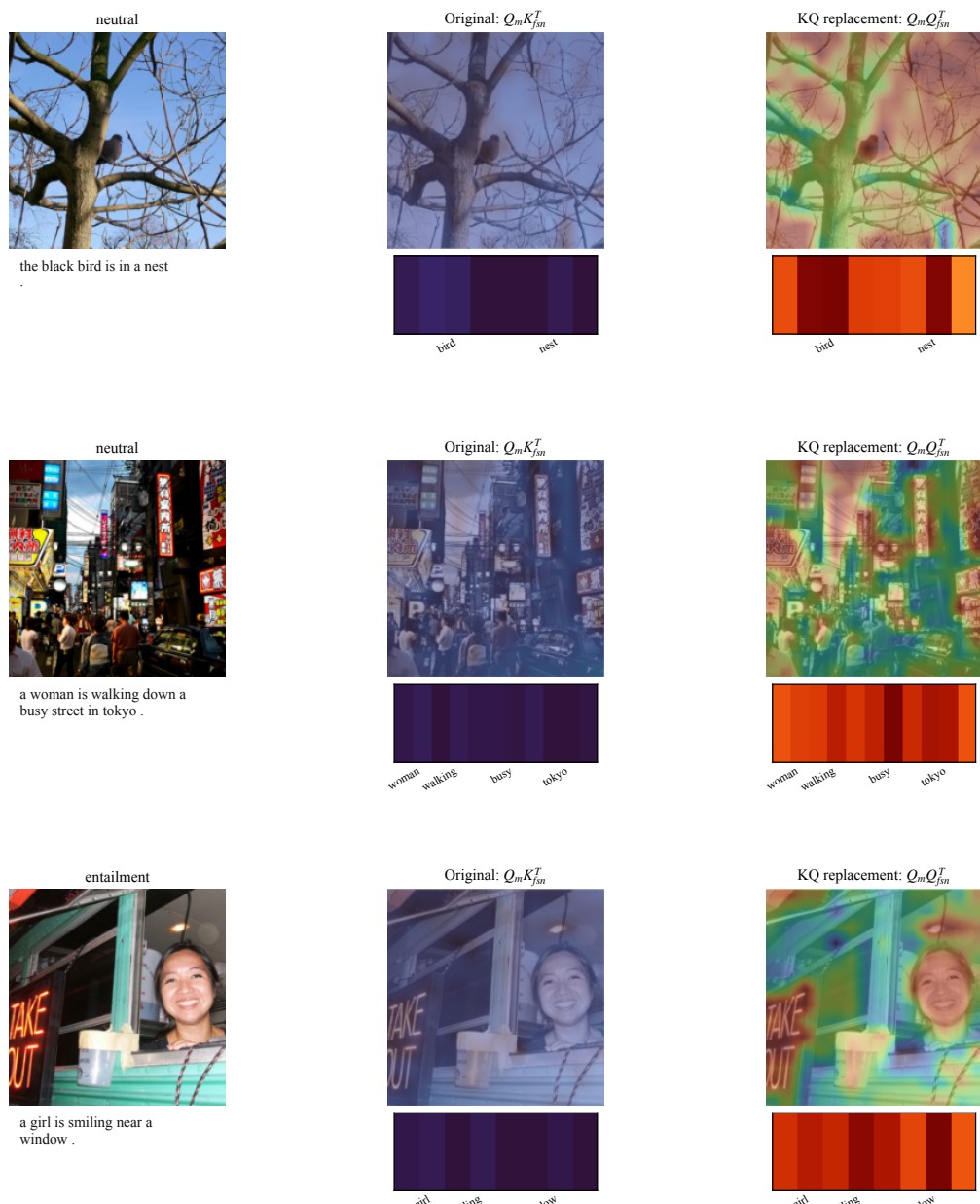

Figure 6: Attention heatmap: Visualization of modality-to-prompt interactions on SNLI-VE. Similar to Food-101 and MM-IMDB, replacing $\mathbf{K}_{\text{fsn}}$ with $\mathbf{Q}_{\text{fsn}}$ leads to stronger initial matching between modality tokens and bottleneck prompt tokens.

by the fusion module. Per fusion layer, we introduce at most $a$ modality-agnostic fusion tokens (introduced only in the the fusion layer), adding $ad$ trainable parameters, and apply LoRA to the value projection $W_v$ with rank $r$, adding $2rd$ parameters ($r \ll d$). The per-layer trainable complexity is therefore $\mathcal{O}(d(a + 2r))$, which is linear in the number of fusion tokens and the LoRA rank.

In comparison, PMF allocates $6a$ prompt tokens per layer (instance/domain/category $\times$ vision/text) and projects them with two-layer MLPs, yielding approximately $\mathcal{O}(6ad + 2ad^2)$ parameters per layer; the $d^2$ term dominates in practice. While this is not a full algorithmic time analysis, the parameter-level characterization

Table 7: Average per-batch latency on MM-IMDB (batch size = 32, averaged over 20 batches).

| Method | Inference Time (ms) |
|---|---|
| PMF | $291.15 \pm 0.12$ |
| Ours | $173.72 \pm 0.11$ |

Table 8: Effect of unified LoRA rank $r^l$ on MM-IMDB with frozen encoders. Higher ranks bring no accuracy gains while increasing trainable parameters. While results computed on a single NVIDIA RTX4090 (24 GB), trends are consistent with the main table. $|\theta|$ denotes the size of trainable parameters. Parentheses (% total) denote the the percentage of total original model parameters.

|  | $|\theta|$ (% total) | MM-IMDB F1-Micro/Macro (%) |
|---|---|---|
| $r^l=8$ | 0.24M (0.12%) | **68.09/60.92** |
| $r^l=16$ | 0.43M (0.22%) | 68.06/60.61 |
| $r^l=32$ | 0.82M (0.42%) | 68.08/60.73 |
| $r^l=64$ | 1.61M (0.82%) | 67.89/60.70 |

tracks the incremental compute/memory of the fusion block and aligns with wall-clock latency: as shown in Table 7, on MM-IMDB under identical settings (batch size = 32, 20 batches), our model averages $173.72\pm0.11$ ms per batch versus $291.15 \pm 0.12$ ms for PMF (an $\sim40.3\%$ reduction).

### A.7 Larger LoRA ranks do not necessarily improve performance.

We unify the LoRA rank across all modules and sweep $r^l \in \{8, 16, 32, 64\}$ on MM-IMDB (Table 8). While increasing $r^l$ enlarges the representational capacity, the gains saturate: both of the best F1-Micro and F1-Macro are achieved at $r^l=8$ (68.09% & 60.92%). In contrast, the trainable parameter count grows substantially (approximately linearly) with $r^l$ from 0.24M to 1.61M, yielding a poor performance at higher ranks. These findings are consistent with the classical conclusions in the LoRA studies.

## B  Discussion: Extension to Contemporary Vision–Language Adaptation

Although our experiments focus on frozen BERT/ViT encoders, the broader design question remains relevant to recent vision–language adaptation: when most pretrained components are frozen or only partially updated, where should limited trainable capacity be placed? Recent work on multimodal instruction tuning has also studied large multimodal models from a representation view, showing that directly editing multimodal representations can improve efficiency and controllability (Liu et al., 2025a). Similarly, recent interleaved vision–language generalists introduce modality-specific trainable components and LoRA variants to better handle heterogeneous visual and textual representations (Xu et al., 2025). These studies suggest that efficient multimodal adaptation is not only a question of reducing parameter count, but also a question of which internal representations are modified.

Our analysis provides a potential attention-level criterion for this question in fusion modules. The value projection determines the token content aggregated by attention, while query–key matching determines how original modality tokens interact with newly introduced fusion tokens. Therefore, when lightweight adaptation is inserted before or inside multimodal fusion, it is useful to ask whether the trainable components directly affect token content or token matching, rather than only increasing the number or complexity of prompt tokens. This view is also consistent with recent PEFT work on pretrained vision–language models, where cross-modal interaction and modality-specific adaptation remain important (Zhao et al., 2025).

## C  Technical Details of Transformers and Attention Mechanisms

### C.1  Complete formulations for multimodal transformers

We consider the pre-trained transformers for two modalities, *i.e.*, text and image in our context. Specifically, the two transformers are the text transformer (*e.g.*, BERT; Devlin et al., 2019) and the vision transformer (ViT; Dosovitskiy, 2020). For simplicity, we consider applying the multimodal transformers only to classification tasks in this formulation.

For the input format, we start at the lowest level and denote a sequence of text input IDs $\mathbf{x}_{\text{txt}} = [x_1, x_2, \ldots, x_N]_{\text{txt}}$ with natural numbers $x_i \in \mathbb{N}$, and an $H \times W$ RGB image $\mathbf{x}_{\text{img}} = [x_1, x_2, \ldots, x_N]_{\text{img}} \in \mathbb{R}^{H \times W \times 3}$, with $x_i \in \mathbb{R}^{h \times w \times 3}$ denotes $i$-th patch within the image and $h \times w$ represents the patch size. (Note here we assume both modalities process equal number of $N$ tokens for notational convenience, while in practice they are unnecessarily the same.)

These input sequences are converted into a series of 1D tokens $z_i \in \mathbb{R}^d$ as follows

$$\mathbf{z} = [z_{\text{cls}}, \mathbf{E}(x_1), \mathbf{E}(x_2), \ldots, \mathbf{E}(x_N)], \tag{7}$$

where $\mathbf{E}$ is a mapping to $\mathbb{R}^d$ which is a patch embedding function $\mathbb{R}^{h \times w \times 3} \to \mathbb{R}^d$ for image inputs and a vocabulary embedding function $\mathbb{N} \to \mathbb{R}^d$ for text inputs, $z_{\text{cls}}$ is a special classification token prepended to the sequence. Positional embedding is omitted here due to irrelevance to the topic.

These tokens are then proceeded into $L$ transformer layers (both practically and for notational simplicity, we assume both transformers have $L$ layers), each consisting of a multi-head self-attention (MHA) module and an MLP block. Reloading the input sequence defined in Equation (7) as $\mathbf{z}^0$, we can repeatedly compute the output of the $l + 1$-th transformer layer $\mathbf{z}^{l+1} = \text{Transformer}(\mathbf{z}^l)$ as

$$\begin{aligned}
\mathbf{y}^{l+1} &= \text{LN}\left(\text{MHA}^{l+1}(\mathbf{z}^l)\mathbf{W}_O^{l+1}\right) + \mathbf{z}^l, \\
\mathbf{z}^{l+1} &= \text{MLP}^{l+1}(\mathbf{y}^{l+1}) + \mathbf{y}^{l+1},
\end{aligned} \tag{8}$$

where LN denotes the layer normalization; (for simplicity) we illustrate the mechanism of $\text{MHA}^{l+1}(\mathbf{z})$ in a single-head form, which is

$$\text{Attn}\left(\mathbf{z}\mathbf{W}_Q^{l+1}, \mathbf{z}\mathbf{W}_K^{l+1}, \mathbf{z}\mathbf{W}_V^{l+1}\right),$$

where $\text{Attn}(\mathbf{Q}, \mathbf{K}, \mathbf{V}) = \text{softmax}(\mathbf{Q}\mathbf{K}^\top/\sqrt{d})\mathbf{V}$ with sets of trainable parameters $\mathbf{W}_{\{Q,K,V,O\}} \in \mathbb{R}^{d \times d}$. The above procedure is performed in both modalities. The final prediction is calculated by the averaged output associated with the two classification tokens from the two modalities.

### C.2  A nonparametric view of attention and the importance of the value part

Previous works (Choromanski et al., 2021; Peng et al., 2021; Chen et al., 2021; 2022a;b) have established a connection between attention and nonparametric estimators, along with several emerging properties inspired from kernel learning. Let us consider the Nadaraya-Watson kernel estimator (Wasserman, 2006, Definition 5.39), which reads

$$f(x) = \sum_{i=1}^n \phi_i(x)Y_i \quad \text{with } \phi_i(x) = \frac{\kappa(x, k_i)}{\sum_{j=1}^n \kappa(x, k_j)}, \tag{9}$$

where $\kappa(\cdot, \cdot)$ is a kernel function, $\{Y_i\}_{i=1}^n$ are the learnable coefficients, and $\{k_i\}_{i=1}^n$ are the supporting points. Consider the kernel function $\kappa(x, y) = \exp(\langle x, y \rangle / \sqrt{d})$, with a slight abuse of notation to extend to matrix/vector form, we have

$$f(\mathbf{z}) = \text{softmax}\left(\frac{\mathbf{Q}\mathbf{K}^\top}{\sqrt{d}}\right)\mathbf{V},$$

$$\text{with } \{\mathbf{Q}, \mathbf{K}, \mathbf{V}\} := \mathbf{z}\mathbf{W}_{\{Q,K,V\}},$$

which recovers the attention operations. With the analogy between attention operations and kernel estimators defined above, previous works (Chen et al., 2022a;b) have summarized emerging guidelines from kernel learning to train attention modules more effectively. Specifically, the representer theorem (Schölkopf et al., 2001) states that the minimizer $f^*(\cdot)$ of some empirical risks admits the representation of the following form

$$f^*(\cdot) = \sum_{i=1}^{n} \alpha_i \kappa(\cdot, k_i),$$

where $\{\alpha_i\}_{i=1}^{n}$ are sets of parameters to optimize.

We can relate this result to the Nadaraya-Watson kernel estimator in Equation (9) by setting $\alpha_i = Y_i / \sum_{j=1}^{n} \kappa(\cdot, k_j)$ to make analogous statements in the context of training attention modules. A useful lecture is, the adaptation to the query–key interactions (*i.e.*, the gram matrix $\mathbf{QK}^\top$) in prompting can be rather conservative, and more computational budgets can be reserved for learning the value $\mathbf{V}$.

