# OpenReview forum: "Simple is Better than Complex: A Representation-centric Perspective for Prompting-based Vision--Language Fusion"
_TMLR — Decision pending for TMLR_

### Review · Reviewer_Ssi2 · 2026-05-01

**Summary Of Contributions:**

This paper studies prompting-based vision-language fusion with frozen unimodal transformers from a representation-centric perspective. The authors argue that increasingly complex prompt designs have limited ability to directly modify internal attention representations, especially the value pathway and intra-modal token interactions. Motivated by this analysis, the paper proposes a simple parameter-efficient fusion method built on attention bottlenecks: value-only LoRA adaptation in selected transformer layers, combined with a key-query replacement strategy for prompt/bottleneck tokens. The method is evaluated on MM-IMDB, UPMC Food-101, and SNLI-VE, where it outperforms several prompting-based baselines, including PMF and a reproduced ViT-based MoPE, while using fewer trainable parameters. The paper also includes ablations on value adaptation, KQ replacement, LoRA rank, larger backbones, latency, and statistical testing.

**Audience:**

Yes

**Audience Explanation:**

The topic is relevant to TMLR's audience. Parameter-efficient vision-language fusion with frozen unimodal backbones is an important and practical problem, especially when full multimodal fine-tuning is expensive. The paper’s central idea—that prompt complexity may be less important than adapting targeted internal attention representations—is interesting and could be useful to researchers working on multimodal learning, PEFT, prompt tuning, and efficient adaptation of pretrained transformers.

**Broader Impact Concerns:**

I do not see major ethical concerns specific to the proposed method beyond the usual concerns for multimodal classifiers and pretrained vision-language systems. The method is parameter-efficient and may reduce the cost of adapting large pretrained models, which is a positive practical impact.

**Claims And Evidence:**

No

**Claims Explanation:**

The empirical results support a narrower claim: the proposed value-LoRA plus KQ replacement method improves over the selected prompting-based fusion baselines on three standard image-text benchmarks, with fewer trainable parameters than PMF and MoPE in the reported setting. The main tables are clear, and the ablations on MM-IMDB suggest that value adaptation is the major contributor and that KQ replacement provides an additional gain.

However, the broader claims are not yet fully supported. First, the paper positions the work as improving prompting-based vision-language fusion, but the proposed method trains LoRA modules in the unimodal encoders. This makes the main comparison partly apples-to-oranges: the method is a PEFT method added to a bottleneck prompt architecture, while many baselines are prompt-only methods. The paper should compare more directly against standard PEFT alternatives, such as LoRA on Q/K/V/O, LoRA on all attention projections, adapter-based methods, IA3-style tuning, and LoRA combined with late fusion or PMF-style prompts. The Pure-LoRA comparison is helpful, but it is only reported on MM-IMDB and does not establish the relative benefit across the main benchmarks.

**Requested Changes:**

1. Add stronger PEFT baselines. Since the proposed method trains LoRA modules, the paper should compare against more direct PEFT baselines, not only prompt-tuning baselines. At minimum, please include LoRA on Q, K, V, O, LoRA on all attention projections, LoRA on MLP layers, adapter-based PEFT, and LoRA plus late fusion or bottleneck fusion. These comparisons should be run on all three main datasets, not only MM-IMDB.
2. Clarify the scope of the claims. The paper should avoid implying that the results prove prompt complexity is generally inferior to representation-centric adaptation. The current evidence supports the more specific claim that value-LoRA plus KQ replacement improves over the selected prompt-based baselines under the tested settings.
3. Strengthen the evidence for KQ replacement. The KQ replacement is interesting but currently heuristic. Please provide ablations on all three datasets, attention-map or representation-similarity analyses showing the claimed alignment effect, and comparisons against simple alternatives such as learning K for prompt tokens, using shared K/Q projections only for prompts, or applying LoRA to K instead.

---

> ### Author Response · Authors · 2026-05-27
> **Response to Reviewer Ssi2**
>
> We thank the reviewer for the detailed assessment. We address the main concerns below.
>
> ### RC1. Stronger PEFT baselines
>
> We add the requested **PEFT baselines** in both Tables 3 and 4. In Table 3, we newly implement ` Q/K/V/O-LoRA` (LoRA applied to each single attention projection separately), `Q,K,V,O-LoRA` (LoRA on all attention projections), `FFN-LoRA` (LoRA on MLP layers), and `Adapter` for the requested "LoRA plus late fusion" baselines.
> > We recall the average PMF performance is provided by the original paper.
>
> | Method | Params | MM-IMDB | Food-101 | SNLI-VE |
> |---|---:|---:|---:|---:|
> | PMF | 2.5M | 64.51/58.77 | 91.51 | 71.92 |
> | $Q$ LoRA | 0.20M | 65.77/57.02 | 92.79 | 68.26 |
> | $K$ LoRA | 0.20M | 66.00/57.80 | 92.81 | 68.49 |
> | $O$ LoRA | 0.20M | 66.81/58.48 | 92.88 | 68.85 |
> | $V$ LoRA | 0.20M | 66.98/59.14 | 93.10 | 69.04 |
> | $Q,K,V,O$ LoRA | 0.20M | 66.47/58.73 | 92.87 | 69.09 |
> | FFN LoRA | 0.25M | 64.68/56.23 | 92.77 | 69.20 |
> | Adapter | 0.20M | 65.49/58.04 | 92.98 | 69.30 |
> | Ours | 0.20M | **68.09/60.92** | **93.53** | **72.31** |
>
>
> These results now better establish two points.
> - First, late-fusion PEFT variants are strong baselines: several PEFT variants already improve over PMF on most datasets while using fewer trainable parameters, which supports the paper's motivation that direct representation adaptation is important.
> - Second, our full method further improves over all the late-fusion PEFT baselines. This supports our design choice.
>
>
> For "LoRA plus bottleneck fusion" baselines, in Table 4 we add 'P-MBT + K-LoRA' and 'P-MBT + V, K-LoRA'. The results are deferred to RC3.
>
>
>
>
> ### RC2. Clarifying the scope of the claims
>
> We have accordingly revised the abstract, introduction, and conclusion to avoid implying that prompt complexity is generally inferior in all general multimodal settings. The current claim is narrower: in the tested frozen BERT/ViT fusion setting, our fusion method improves over the compared prompting-based and PEFT baselines.
>
>
> ### RC3. Evidence for KQ replacement
>
> We appreciate this suggestion and add new evidence for KQ replacement. We first recall the mechanism. KQ replacement only changes the **modality-to-prompt block** at the initial bottleneck interaction stage: the original matching term $Q_m K_{fsn}^T$ is replaced by $Q_m Q_{fsn}^T$. This zero-parameter change does not modify value aggregation; it changes how newly initialized prompt tokens are matched by pretrained modality queries.
>
>
>
> 1. Appendix A.5 adds initialization-stage **attention maps** on MM-IMDB, Food-101 and SNLI-VE. These figures visualize the exact modality-to-prompt block changed by KQ replacement: the original model uses $Q_m K_{fsn}^T$, while KQ replacement uses $Q_m Q_{fsn}^T$. The visualization is not a full self-attention matrix and not attention maps from a trained model; each heatmap value is a pre-softmax score averaged over 12 heads and 4 prompt tokens.
>
>     Across the three datasets, $Q_m Q_{fsn}^T$ produces stronger modality-to-prompt responses than $Q_m K_{fsn}^T$ at initialization. This visualization supports the design motivation in Section 4.4: the newly initialized prompt tokens better match pretrained modality queries and obtain higher attention scores, making the bottleneck prompt tokens more actively trained.
>
>
> 2. In Table 4, we compare KQ replacement with learning `W_K` itself:
>
>     | Method | Purpose | Params | MM-IMDB | Food-101 | SNLI-VE |
>     |---|---|---:|---:|---:|---:|
>     | P-MBT | bottleneck baseline | <0.1M | 64.86/56.45 | 89.88 | 62.79 |
>     | P-MBT + K-LoRA | learning `W_K` alone | 0.20M | 66.31/57.77 | 92.84 | 70.58 |
>     | P-MBT + V-LoRA | value adaptation baseline | 0.20M | 67.34/59.32 | 93.13 | 71.52 |
>     | P-MBT + V, K-LoRA | value adaptation + additionally learning `W_K` | 0.25M | 67.53/60.29 | 93.29 | 71.64 |
>     | Ours: P-MBT + V-LoRA + KQ replacement | our zero-extra-parameter design | 0.20M | **68.09/60.92** | **93.53** | **72.31** |
>
>     (For `P-MBT + V, K-LoRA`, the V-LoRA rank is unchanged and extra rank-2 K-LoRA is added. This variant uses more trainable parameters than the KQ replacement variant.)
>
>     We study three suggested alternatives as follows:
>
>     - For "learning K for prompt tokens", `P-MBT + V, K-LoRA` uses extra K-LoRA to align the key prompt token with the query modality token. It does bring higher performance than the `P-MBT + V-LoRA` baseline at a cost of more trainable parameters. However, we note it is no better than our proposed full approach.
>
>     - For "using shared K/Q projections only for prompts", we note our KQ replacement technique already follows this idea. As shown in Figure 2, we keep using $K_m$ for modality tokens, while enforcing shared K/Q projections $K_{fsn} = Q_{fsn}$ only for fusion prompts.
>
>     - For "applying LoRA to K instead", we implement `P-MBT + K-LoRA`. Given the limited parameter budget, this alternative sacrifices value adaptation and deviates from our analysis in Section 4.3.

---

### Review · Reviewer_xNNg · 2026-05-09

**Summary Of Contributions:**

The paper studies parameter-efficient fusion of two separately pre-trained unimodal transformers (e.g., ViT for image, BERT for text) for classification from joint (vision, language) inputs, in the "interactive prompting" line of work. The core argument is that prompt-only designs face a structural limitation on their ability to steer the modalities' internal representations, and that targeted edits to the attention mechanism are a more effective approach.

The authors give a decomposition of the attention mechanism between prompt and non-prompt positions, showing structural limitations on what the prompt-tokens can do to the outputs of the attention mechanism. They propose two modifications to the traditional approach, firstly adding a LoRA to the value heads, and secondly re-organizing the attention matrix to switch the query and key when dealing with the fusion tokens. Empirical results show that the approach gets increased scores on the relevant evaluations with fewer parameters than the baselines, and give ablations showing the relative importance of each of the components.

**Audience:**

Yes

**Audience Explanation:**

Although I am not an expert in this field, I understand that the paper sits in the relatively niche area of parameter-efficient multimodal fusion via frozen unimodal encoders. The comparison points cited by the papers, such as MoPE and PMF, are peer-reviewed and relatively recent, so I assume this contribution will be of interest to that subfield. I was not able to find any work work that is precisely the same as this, so I think it is genuinely novel and would be of interest. I think the audience will also be interested in the attention decomposition, as it is a structural limitation that is common to all methods in the parameter efficient multimodal fusion area. While I understand that most work in multimodal approaches tends to use VLMs nowadays, I think this work would still be of interest.

**Claims And Evidence:**

Yes

**Claims Explanation:**

Most claims are well supported. The basic idea is that targeted LoRA adapters in key places in the network can improve performance with fewer parameters than heavy optimization of the prompt tokens, and this is upheld by the experiments and motivated well by the theoretical arguments. The empirical comparisons in Table 2 are clear and hold up under significance testing.

The paper occasionally makes some implicit over-claims, such as the importance of LoRA vs the attention decomposition, which I discuss in the requested changes section. I don't think these are critical issues and can be fixed with changing some wording/re-organization.

**Requested Changes:**

# Strengthen The Work
## Over-Claiming
I think the paper tends to over-state some claims. There are two main areas where this happens. I think both can be addressed with changes to the framing/wording and don't require any additional experiments. Firstly, the narrative emphasis on representation-centric adaptation is perhaps stronger than the ablation results justify. The abstract, introduction, and conclusion all focus on representation-centric adaptation as the primary lever, with the lengthy derivation of the attention structure in the middle. Meanwhile, the ablations show that simple LoRA adapters in either the 'Pure-LoRA' setting or V-only give very strong results, stronger than PMF. I feel like the paper could be summarized with 'LoRA adapters are effective at fine-tuning models with small numbers of parameters' which is not especially novel. It also makes me wonder how effective it would be to simply have adapters on K,Q as well. It could be more honest to present the work as 'LoRA adapters are effective in this task, we conduct analysis to design especially-effective low-parameter adapters and attention modifications for fusion.'

Secondly, the analysis in section 4.1 is used to support multi-layer claims. Section 4.1 derives the structural limitation in a single layer, and says that this might not apply to multi-layer fusion. However, the claims in the abstract, e.g. "interactive prompting itself has limited ability to directly alter value token representations and intra-modal token interactions" use this single-layer result as a more general claim about prompt-tuning's limits across the full network. It would be more honest to either continue this analysis across multiple layers, give examples showing that single-layer fusion is very common, or make the claims a bit less broad.

## More details on Pure-LoRA
As far as I understand, Pure-LoRA is the cleanest baseline for the paper's argument about the importance of representation-centric adaptation, and (I think) is a very informative comparison. It currently appears only in the ablation section. There are also not many descriptions about what exactly it is. There should be more details and it should be more prominently featured.

## Cross-Dataset Ablations
The ablations are a bit limited currently, since they are only on MM-IMDB. It would be very useful to have at least one more dataset, to see if the relative benefits of each component are of the same order, or if it's dataset-dependent.

## More Motivation on Benefits of Parameter-Efficiency
As a relative outsider to the field, it is not especially clear why fewer trainable parameters are necessarily better than more, especially when comparing e.g. 0.5 million vs 5 million or even 200 million parameters. Both 0.5 and 200 million trainable parameters will easily fit and train in any modern GPU. Similarly, both the fusion and methods with more parameters, such as ViT, still take the same compute to do inference, since the fusion method has the ViT as part of it. LoRA adapters can also be merged-in with no inference overhead. As far as I can tell, the implicit claim is that smaller numbers of parameters are less prone to overfitting on the small datasets--I think it would be very nice to have a direct analysis of this, e.g. test accuracy as a function of number of training examples, instead of just promoting the proxy of number of parameters.

---

> ### Author Response · Authors · 2026-05-27
> **Response to Reviewer xNNg**
>
> We thank the reviewer for the thoughtful comments. We address the main concerns below.
>
> ### RC1.1 the narrative emphasis on representation-centric adaptation
>
> We respectfully recall that our method is **motivated** by a representation-centric perspective, which is also highlighted in Section 4.3. In our paper, LoRA is used as a lightweight tool to instantiate representation adaptation inside attention. We **have revised the abstract, introduction, and conclusion** to state this more precisely: in frozen vision--language fusion, LoRA provides a flexible way to **adapt attention representations**, while our representation-centric analysis guides the design of low-parameter attention adaptation and fusion modifications.
>
> Moreover, after adding new PEFT baselines, the revised Table 3 better supports this framing. These baselines confirm that representation-centric adaptation (not limited to LoRA) is strong, and our full method even significantly outperforms the strongest late-fusion PEFT variant, showing that the overall gain from adopting the representation-centric perspective is not explained by LoRA alone. Thus, in the revision we present LoRA as a tool for representation adaptation, with the main motivation remaining the representation-centric analysis.
>
>
>
>
> ### RC1.2 Section 4.1 and multi-layer prompting
>
> We agree that the summary wording can better align with the actual analysis in Section 4.1. As implied in our paper, we acknowledge that our per-layer analysis does not preclude multi-layer effects, since prompt tokens can influence later-layer inputs through earlier-layer outputs. However, inside each frozen layer, the value projection itself remains fixed unless it is directly adapted.
>
> We thank the reviewers for pointing this out and have revised the abstract, introduction, and conclusion accordingly. We have marked the revisions in blue to ensure that the revised wording does not exceed the scope of Section 4.1.
>
>
> ### RC2. More details on Pure-LoRA
>
> We appreciate this suggestion and indeed expand the previous Pure-LoRA reference into late-fusion PEFT variants in the revised `Table 3`, including LoRA on $Q/K/V/O$, LoRA on all attention projections ($Q,K,V,O$ LoRA), LoRA on Transformer FFN layers, and adapter-based PEFT. We add clear descriptions of new baselines in Appendix A.4.
>
> ### RC3. Cross-dataset ablations
>
> We add Food-101/SNLI-VE results for the late-fusion PEFT variants in `Table 3` and add Food-101/SNLI-VE results for the bottleneck fusion variants in `Table 4`. The trend is consistent with MM-IMDB.
>
> ### RC4. More Motivation on Benefits of Parameter efficiency
>
>
> We lay out the benefits of parameter efficiency in this response. We hope to indicate this is a cliche in this field and thus not the main topic in our paper. Some of the points were already contained in the `Appendices A.6-A.7`, and we make the evidence more visible in the revision.
>
> - One direct benefit is that users can significantly reduce GPU memory usage **in training**, which matters since training a full-parameter model with Adam will take four times as much memory as inference. In particular, we note it is struggling for users to train a 200M full-parameter FP32 model on a 16G V100 GPU (a "modern GPU").
>
> - Also, in introducing more parameter-efficient adaptation designs, we further reduce the inference runtime compared to **previous prompting-based methods**. Specifically, `Table 7` already reported $173.72±0.11$ ms per batch for our method versus $291.15±0.12$ ms for PMF on MM-IMDB, suggesting a reduction of about 40.3%.
>
> - For test accuracy and trainable parameter counts, `Table 8` already showed that lifting the LoRA rank from 8 to 64 increases trainable parameters while not necessarily improving performance.

---

### Review · Reviewer_zXgG · 2026-05-15

**Summary Of Contributions:**

This paper studies multimodal fusion from a representation-centric perspective. The authors argue that existing prompt-based multimodal fusion methods primarily manipulate prompt tokens while leaving the underlying attention value representations and intra-modal interactions largely unchanged. This paper proposes a lightweight alternative that directly adapts attention internals instead of introducing increasingly sophisticated prompt architectures. Concretely, the method combines value-only LoRA adaptation on attention value projections and a lightweight “key-query replacement” mechanism. This mechanism projects prompt tokens using the query projection instead of the key projection, reducing representation mismatch between pretrained modality tokens and newly introduced prompt tokens during attention computation. The approach is evaluated on MM-IMDB, UPMC Food-101, and SNLI-VE, where it consistently outperforms prior prompt-based fusion baselines while using fewer trainable parameters than competitive methods. The paper also includes several ablation studies supporting the importance of value adaptation and the proposed KQ replacement mechanism.

**Audience:**

Yes

**Audience Explanation:**

Yes this methodology provides increased performance relative to other prompt-based methods. Adapting models with PEFT is an area of interest to many in the TMLR community and this paper provides a clever method to improve performance. The motivation of and presentation of the heuristics would also be of interest.

**Claims And Evidence:**

Yes

**Claims Explanation:**

The empirical evidence is generally strong and supports the main practical claims of the paper. The authors validate on multiple datasets against a variety of baselines, including both prompt-based methods and fine-tuning. The proposed method consistently improves over prior prompting-based fusion baselines across multiple benchmarks while remaining competitive with fine-tuning despite using substantially fewer trainable parameters. The accompanying ablation studies help isolate the contribution of each proposed component. The progression from P-MBT -> V-LoRA -> V-LoRA + KQ  directly supports the paper’s central argument that representation-level adaptation inside attention is more effective than increasing prompt complexity alone.

The conceptual claims are supported more by intuition than by rigorous theoretical evidence. The discussion regarding the inherent limitations of prompt-based fusion and the relative importance of value representations versus query/key interactions is plausible and well-motivated, but remains largely heuristic.

**Requested Changes:**

Both requested changes would strengthen the work, but are not critical.

1. While the ablations are strong and thoughtfully designed, the paper could be further strengthened by including comparisons against a broader set of PEFT baselines to better isolate the contribution of the proposed design itself.

2. A brief discussion or small-scale experiment on more modern multimodal architectures (e.g., CLIP-style or instruction-tuned vision-language models) would strengthen the paper’s broader relevance. The current experiments are well-executed within the chosen BERT/ViT fusion setting, but some commentary on how the proposed ideas may extend to contemporary multimodal systems would make the paper more compelling to a wider audience.

---

> ### Author Response · Authors · 2026-05-27
> **Response to Reviewer zXgG**
>
> We sincerely thank the reviewer for the positive assessment and constructive suggestions. We appreciate the recognition of the paper's motivation and empirical evaluation. Below we address the two requested additions in detail.
>
>
> ### RC1. Broader PEFT baselines
>
> We add the requested PEFT baselines in revised `Table 3`. The table keeps the original role of this ablation: it separates prompt-based fusion, representation adaptation without explicit cross-modal fusion before the classifier, and our full fusion design. Concretely, we expand the previous Pure-LoRA reference into late-fusion PEFT variants, including LoRA on $Q/K/V/O$, LoRA on all attention projections, LoRA on Transformer FFN layers, and adapter-based PEFT.
> > We recall the average PMF performance is provided by the original paper.
>
> | Method | Params | MM-IMDB | Food-101 | SNLI-VE |
> |---|---:|---:|---:|---:|
> | PMF | 2.5M | 64.51/58.77 | 91.51 | 71.92 |
> | $Q$ LoRA | 0.20M | 65.77/57.02 | 92.79 | 68.26 |
> | $K$ LoRA | 0.20M | 66.00/57.80 | 92.81 | 68.49 |
> | $O$ LoRA | 0.20M | 66.81/58.48 | 92.88 | 68.85 |
> | $V$ LoRA | 0.20M | 66.98/59.14 | 93.10 | 69.04 |
> | $Q,K,V,O$ LoRA | 0.20M | 66.47/58.73 | 92.87 | 69.09 |
> | FFN LoRA | 0.25M | 64.68/56.23 | 92.77 | 69.20 |
> | Adapter | 0.20M | 65.49/58.04 | 92.98 | 69.30 |
> | Ours | 0.20M | **68.09/60.92** | **93.53** | **72.31** |
>
> These results now better establish two points.
> - First, late-fusion PEFT variants are strong baselines: some PEFT variants already improve over PMF on most datasets while using fewer trainable parameters, which supports the paper's motivation that direct representation adaptation is important.
> - Second, our full method further improves over all the late-fusion PEFT baselines. This supports our design choice.
>
>
>
> ### RC2. Relation to CLIP-style or instruction-tuned VLMs
>
>
> We thank the reviewers for encouraging us to clarify the connection to contemporary multimodal systems. While our experiments focus on frozen BERT/ViT fusion, similar design questions may arise in modern VLM adaptation: where should limited trainable capacity be placed when pretrained multimodal components are frozen or only partially updated? Recent studies approach related questions through representation-level multimodal instruction tuning [1], modality-specialized adaptation for interleaved vision-language generalists [2], and low-rank adaptation for missing-modality visual recognition [3].
>
> In this context, our analysis provides a representation-centric design implication for attention-based fusion. The value projection controls the token content aggregated by attention, while query--key matching controls how modality tokens interact with newly introduced fusion tokens. This perspective suggests that future lightweight adaptation for contemporary VLMs may benefit from targeting internal representations that directly affect token content or token matching, rather than only increasing prompt complexity.
>
> We have included a discussion of this potential as a design implication in **Appendix B** of the revised edition.
>
> ### References
>  [1] Liu et al., "Re-Imagining Multimodal Instruction Tuning: A Representation View", ICLR, 2025.
>  [2] Xu et al., "Modality-Specialized Synergizers for Interleaved Vision-Language Generalists", ICLR, 2025.
>  [3] Zhao et al., "MoRA: Missing Modality Low-Rank Adaptation for Visual Recognition", ICLR, 2026.

---

### Author Response · Authors · 2026-05-27
**General Response**

Dear Editor and Reviewers,

We sincerely thank the Action Editor and all reviewers (`zXgG`, `xNNg`, and `Ssi2`) for the careful reading and constructive feedback. We appreciate the reviewers' recognition of the motivation, empirical evaluation, and parameter-efficient design of our work.

Our paper studies prompting-based vision--language fusion from a representation-centric perspective. We analyze how prompt tokens interact with frozen attention representations, and instantiate this analysis by adding low-rank adaptation and a lightweight key--query replacement strategy to an attention-bottleneck fusion baseline.

Based on the reviewers' suggestions, we revise the manuscript along multiple directions.

### 1. Clarifying the Scope and Technical Positioning

Reviewers `xNNg` and `Ssi2` noted that the original wording could sound broader than what our analysis and experiments strictly support. We therefore revise the **abstract, introduction, and conclusion** to make the scope more explicit. Our claim is now strictly scoped to the tested frozen-encoder fusion setting: flexible representation adaptation provides a strong alternative to further increasing prompting-architecture complexity.

This revision also clarifies the role of LoRA. LoRA is used as a lightweight tool to instantiate representation adaptation.

### 2. Add stronger PEFT Baselines

In response to `zXgG`, `xNNg`, and `Ssi2`, we expand the original Pure-LoRA reference in **Table 3** into a broader set of late-fusion PEFT baselines, including $Q/K/V/O$-LoRA, LoRA on all attention projection matrices, FFN LoRA, and Adapter.

The new results show that late-fusion PEFT variants are strong baselines: several variants improve over PMF while using fewer trainable parameters. At the same time, our full method improves over all late-fusion PEFT baselines, showing that the final gain is not explained by generic PEFT alone. We add implementation details in **Appendix A.4**.

### 3. Add more evidence for KQ Replacement

Reviewer `Ssi2` asked whether KQ replacement is mainly heuristic and whether directly learning `W_K` could replace it. We clarify that KQ replacement changes only the **modality-to-prompt block**, replacing $Q_m K_{fsn}^T$ with $Q_m Q_{fsn}^T$, without changing modality-token keys or value aggregation.

We add stronger ablations in **Table 4**, including `P-MBT + K LoRA` and `P-MBT + V,K LoRA`. These comparisons show that key-side adaptation helps, but the full method with KQ replacement achieves the best performance without adding trainable parameters beyond the $V$-LoRA variant. We further add attention heatmap visualizations in **Appendix A.5** on all three datasets to show the matching block directly affected by KQ replacement.

We believe these revisions clarify the claim scope, strengthen the empirical comparisons, and provide clearer evidence for the proposed design.

Best regards,
Authors